# Tuning the hysteresis of a metal-insulator transition via lattice compatibility

Y. G. Liang[1,10], S. Lee[1,2,10], H. S. Yu[1], H. R. Zhang [3,4], Y. J. Liang [5], P. Y. Zavalij [6], X. Chen [7], R. D. James[8], L. A. Bendersky[3,4], A. V. Davydov [4], X. H. Zhang [1✉] & I. Takeuchi [1,9✉]

Structural phase transitions serve as the basis for many functional applications including shape memory alloys (SMAs), switches based on metal-insulator transitions (MITs), etc. In such materials, lattice incompatibility between transformed and parent phases often results in a thermal hysteresis, which is intimately tied to degradation of reversibility of the transformation. The non-linear theory of martensite suggests that the hysteresis of a martensitic phase transformation is solely determined by the lattice constants, and the conditions proposed for geometrical compatibility have been successfully applied to minimizing the hysteresis in SMAs. Here, we apply the non-linear theory to a correlated oxide system ($V_{1-x}W_xO_2$), and show that the hysteresis of the MIT in the system can be directly tuned by adjusting the lattice constants of the phases. The results underscore the profound influence structural compatibility has on intrinsic electronic properties, and indicate that the theory provides a universal guidance for optimizing phase transforming materials.

[1] Department of Materials Science and Engineering, University of Maryland, College Park, MD 20742, USA. [2] Department of Physics, Pukyong National University, Busan 48513, South Korea. [3] Theiss Research, Inc, La Jolla, CA 92037, USA. [4] Material Science and Engineering Division, Materials Measurement Laboratory, National Institute of Standards and Technology, Gaithersburg, MD 20899, USA. [5] Chemical and Biomolecular Engineering, University of Maryland, College Park, MD 20742, USA. [6] Department of Chemistry and Biochemistry, University of Maryland, College Park, MD 20742, USA. [7] Department of Mechanical and Aerospace Engineering, Hong Kong University of Science and Technology, Clear Water Bay, Hong Kong. [8] Department of Aerospace Engineering and Mechanics, University of Minnesota, Minneapolis, MN 55455, USA. [9] Maryland Quantum Materials Center, University of Maryland, College Park, MD 20742, USA. [10] These authors contributed equally: Y. G. Liang, S. Lee. ✉email: xhzhang@umd.edu; takeuchi@umd.edu

The hallmark of first-order structural transformations in solid materials are dramatic changes in physical properties with significant technological implications including caloric effects[1], metal-insulator transitions (MITs)[2], and enhanced dielectric/piezoelectric susceptibility[3]. For metallic alloys, lattice compatibility of the parent phase and the product phase at transformation has proven to be a key factor governing the reversibility of the transition as manifested in the hysteresis of the structural transition[4]. Minimization of the hysteresis through tuning of lattice constants, as encoded in the geometrically non-linear theory of martensite[5], has led to development of shape memory alloys with exceptional functional fatigue properties[5–8]. In particular, by adjusting the middle eigenvalue $\lambda_2$ of the transformation stretch tensor (a $3 \times 3$ matrix that describes the structural transformation) to 1, a recipe prescribed by the non-linear theory, a precipitous drop in thermal hysteresis was observed. When more-stringent conditions (the cofactor conditions) are satisfied[8], a shape memory alloy was found to show unusual domain patterns encompassing multiple length scales and reflecting the ultra-compatibility of the martensite and austenite[5].

Given the ubiquitous nature of first-order transformations, it is of interest to explore the applicability of the non-linear theory of martensite to functional oxide materials: can the brittle ceramic materials also be engineered to have highly-reversible transformations through fine-tuning of the lattice constants? In this report, we demonstrate that by tuning the lattice constants of the high-temperature tetragonal phase and the low-temperature monoclinic phase in W substituted $VO_2$, the thermal hysteresis of the MIT can indeed be controlled as the middle eigenvalue of the transformation stretch tensor is changed.

As an archetypical $3d^1$-correlated oxide, vanadium dioxide ($VO_2$) shows a MIT[9] at the transition temperature ($T_C$) of $\approx340$ K. Although $VO_2$ is known to have various polymorphs, the change in the resistivity of the material was found to be intimately associated with a first-order structural phase transformation between a low-temperature monoclinic phase (M1 phase) and a high-temperature rutile-type tetragonal phase (R phase)[10–13]. The relation between the structural phase transformation and the MIT in $VO_2$ has been extensively studied. In particular, many experiments have indicated that the MIT in $VO_2$ is induced by an electron-lattice interaction (i.e., a Peierls transition) through the structural phase transformation[14–16]. However, there have been increasing experimental evidence, suggesting that the resistance switch and the structural phase transformation in $VO_2$ can be decoupled, and thus the MIT is primarily driven by an electron–electron interaction (i.e., a Mott transition)[17–19]. Moreover, in addition to the changes in the structural and the electronic properties, $VO_2$ also shows marked changes in many other properties, e.g., the optical transmittance[20], making the material attractive for a number of practical applications, including smart-window coatings, ultrafast sensors, and switching devices[10–13,20–22]. The thermal hysteresis width of pure $VO_2$ is relatively large (>10 K for a polycrystalline film)[23], which is detrimental to applications requiring agile reversible processes and a large number of reversible cycles.

From the viewpoint of tuning composition to satisfy strong conditions of compatibility between phases, $VO_2$ is an extremely unusual material[8]. To explain this assertion, we first note that in general there are two levels of conditions of compatibility known: (1) $\lambda_2 = 1$ and (2) the cofactor conditions. The first level of these conditions ($\lambda_2 = 1$) is necessary and sufficient that there is a perfect unstressed interface between phases. The second level of these conditions (cofactor conditions) includes $\lambda_2 = 1$, together with another condition associated with minimal volume expansion or stretching through the phase transformation. The cofactor conditions not only imply perfect unstressed interfaces between R and any single variant of M1, but also imply a large number of low energy interfaces with any pair of M1 variants, at any volume fraction. The two known alloys[5,24] to accurately satisfy the cofactor conditions, namely $Zn_{45}Au_{30}Cu_{25}$ and $Ti_{54}Ni_{34}Cu_{12}$, have exceptional reversibility of the transformation, including in one case perfect reversibility after 10 million cycles of full stress-induced transformation, under tension, at peak stresses each cycle of 400 MPa.

The crystallographic specifics of the phase transformation in $VO_2$ are rather rare. Specifically, when $\lambda_2 = 1$ is satisfied, the cofactor conditions are then automatically satisfied[8]. Therefore, satisfying $\lambda_2 = 1$ in this oxide becomes especially important. In particular, the cofactor conditions are satisfied for the compound twins in this material, of which there are many examples, depending on the choices of variants of the M1 phase. Since we satisfy $\lambda_2 = 1$ in this paper to high accuracy, we here add a third member to the list of "cofactor materials".

It is known that the transition properties of $VO_2$ can be effectively tuned through a variety of means, such as chemical substitution, electrical field, optical irradiation, external stress, etc.[10–13,21]. Among these approaches, chemical substitution has been extensively investigated[25–29]. In particular, tungsten (W) has been reported to reduce the transition temperature by ~21–28 K for each atomic percentage of W ions in $VO_2$[28,30]. However, tunability of the thermal hysteresis width upon increasing the W concentration and its mechanism have not been well understood[31]. Through a systematic study using thin-film composition spreads, we show that at $\approx2.5$ at. % substitution, the middle eigenvalue $\lambda_2$ of the transformation stretch tensor becomes 1, and the lattice parameters also satisfy the cofactor conditions, resulting in reduced thermal hysteresis width of the MIT. Our work underscores the inescapable consequence of lattice compatibility, and signals a unique pathway to control functionalities in a variety of materials including strongly correlated electron systems.

## Results

**Film deposition and composition characterization.** In this study, a combinatorial film fabrication strategy[32] was adopted to ensure identical deposition parameters for samples with different W concentrations on a given chip. As illustrated in Fig. 1a, continuous composition-spread films of $V_{1-x}W_xO_2$ ($0 \leq x < 4.0\%$) were fabricated by alternatively ablating a $V_2O_5$ target (A) and a $V_{1.92}W_{0.08}O_2$ target (B) using pulsed-laser beams at an oxygen environment with a pressure of $\approx0.4$ Pa; during the ablation of the two targets, an automated moving mask was used to generate a composition gradient across a $c$-$Al_2O_3$ (0001) substrate (henceforth, denoted as $V_{1-x}W_xO_2/c$-$Al_2O_3$) or across a Si substrate (henceforth denoted as $V_{1-x}W_xO_2$/Si). The highest W concentration ($x$) in these films was controlled to be <4.0% to prevent potential phase separation, which was reported for higher W concentrations[33]. As illustrated in Fig. 1b, each composition-spread film was then patterned into multiple parallel strips (each with a length of 5 mm and a width of 0.2 mm) perpendicular to the composition gradient direction for further measurements. Moreover, according to the change in the composition across the entire substrate and the width of each strip, the variation of the W concentration within each strip is estimated to be ±0.1%. It should be noted that throughout the report, the specific value of the W concentration ($x$) is expressed as a percentage. For example, $x = 2\%$ corresponds to a chemical composition of $V_{1.98}W_{0.02}O_2$.

**Crystal structure characterization by X-ray diffraction.** The X-ray diffraction (XRD) patterns of the sample strips across the

composition-spread films were measured at different temperatures. From the XRD results obtained on both $c$-$Al_2O_3$ and Si substrates, only certain peaks associated with the monoclinic phase and/or the rutile-type tetragonal phase of $V_{1-x}W_xO_2$ could be identified in the entire scan range ($10° \leq 2\theta \leq 90°$), suggesting the absence of other phases of vanadium oxides and tungsten oxides. The XRD results obtained from $V_{1-x}W_xO_2$/Si samples show a number of diffraction peaks revealing the polycrystalline nature of these films. Among the peaks observed in the diffraction scans, the one appearing at ~28.0° shows strong intensity, thus allowing the study of the phase evolution. Figure 2 shows the XRD patterns of a $V_{1-x}W_xO_2$/Si sample in a range from 26° to 31° at different temperatures.

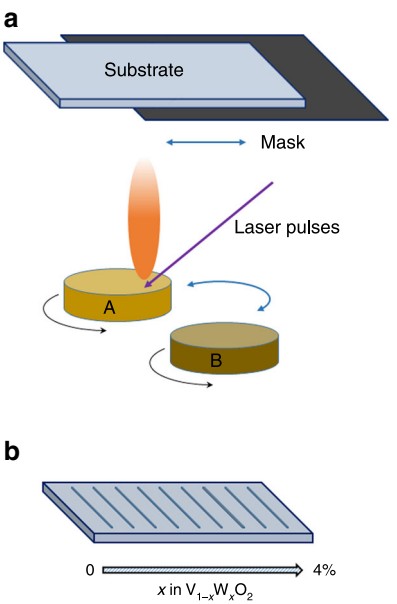

**Fig. 1 Fabrication of a composition-spread film. a** A schematic diagram of the set-up for film growth: in cooperation with the reciprocating movement of a mask, laser pulses are applied to alternately ablate target A and target B to obtain a continuous composition-spread film; **b** a schematic view of a patterned spread film of $V_{1-x}W_xO_2$.

As shown in Fig. 2a, patterns obtained at 300 K from each of the four sample strips with $x < 1.5\%$ (see Materials and Methods) clearly show a peak at around 28.0°, which is consistent with the (011) of the M1 phase; the scans obtained from the strips with $x > 1.5\%$ shows a peak at around 27.75°, which is the (110) of the R phase. Moreover, the pattern obtained from the sample strip with $x = 1.5\%$ shows a relatively broad peak, manifesting the coexistence of the M1 phase (the peak at ~28.00°) and the R phase (the peak at ~27.75°). Therefore, by increasing the W concentration, the insulating M1 phase at 300 K gradually evolves into the metallic R phase. Compared with the slight shift of the peak towards a lower angle within the M1 phase or within the R phase entirely owing to smaller $V^{4+}$ ions replaced by bigger $W^{4+}$ ions[31,34,35], the shift of the diffraction peak is more prominent when the crystal structure changes from the R phase to the M1 phase at a critical W concentration of ~1.5%.

As shown in Fig. 2b, at 323 K, the position of the peak obtained from each sample strip with $x > 1.5\%$ remains unchanged. The peak for the sample strip with $x = 1.5\%$ becomes sharper and clearly centered at ~27.75°, indicating that the sample strip is fully in the metallic R phase. Compared to that observed at 300 K, the peak for the sample strip with $x = 0.9\%$ at 323 K appears at an angle 0.25° smaller, suggesting that a transformation from the M1 phase to the R phase is completed as the temperature increases. Moreover, a clear double-peak feature observed in the pattern obtained from each of the three sample strips with $x < 0.9\%$ indicates the coexistence of the two phases at 323 K.

As shown in Fig. 2c, when the temperature is further increased to 358 K, all the sample strips show a sharp peak at around 27.75° in the XRD patterns. Therefore, with $x$ in the range of 0 to 3.4%, the entire composition-spread $V_{1-x}W_xO_2$ film is in the high-temperature metallic R phase. The slight shift of the peak towards a lower angle as $x$ increases is an indication of the size effect of the substitution of V with W in the $V_{1-x}W_xO_2$ system[31,34,35].

In contrast to the multi-peak polycrystalline XRD results obtained from the $V_{1-x}W_xO_2$/Si sample, the XRD results of all the sample strips in a $V_{1-x}W_xO_2$ film fabricated on a $c$-$Al_2O_3$ substrate show only two peaks belonging to the film at ~40.00° and ~86.30°, respectively (Fig. 3a). The positions of the two peaks correspond to the (020) and the (040) of a slightly W-doped $VO_2$ film in either the M1 phase ($b_{M1} = 4.52$ Å)[36] or the R phase ($a_T = 4.55$ Å)[37]. Therefore, the results suggest that the $V_{1-x}W_xO_2$ film was epitaxially grown on the $c$-$Al_2O_3$ substrate following the

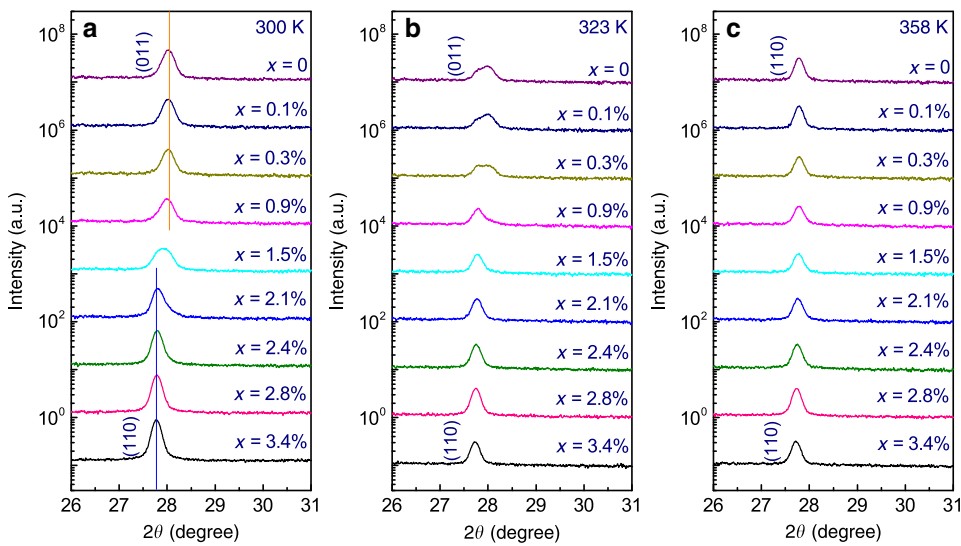

**Fig. 2 XRD results of a $V_{1-x}W_xO_2$/Si film. a–c** $\theta$–$2\theta$ patterns obtained from different sample strips of the $V_{1-x}W_xO_2$/Si film at 300, 323, and 358 K, respectively. An orange line and a blue line are placed in **a** to, respectively, indicate the peak positions in the two-end phases.

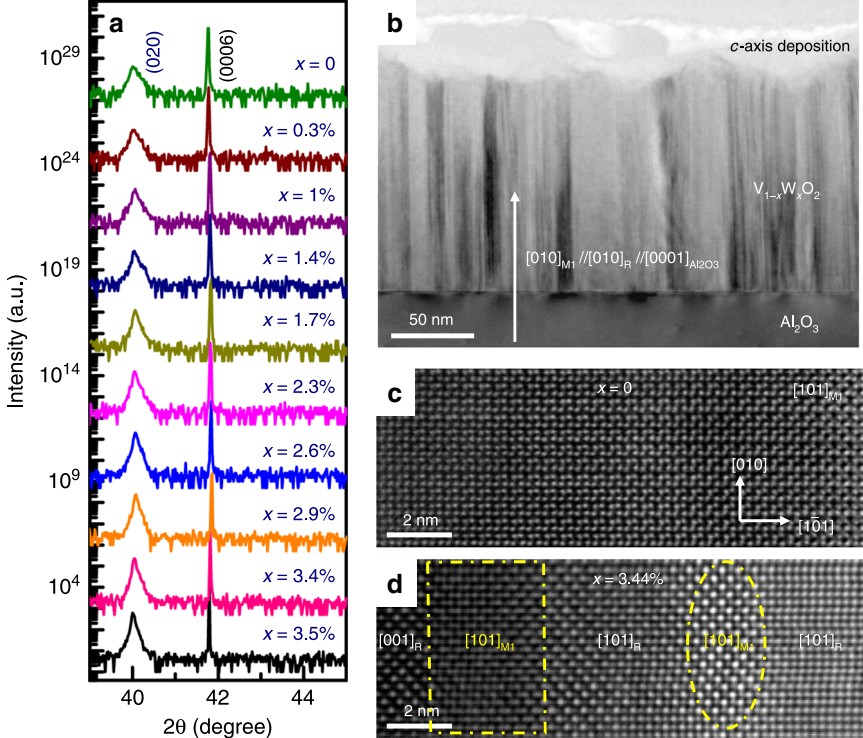

**Fig. 3 XRD and TEM results of a V$_{1-x}$W$_x$O$_2$/c-Al$_2$O$_3$ film. a** XRD θ–2θ patterns obtained from different sample strips of a V$_{1-x}$W$_x$O$_2$ (0 ≤ x < 4.0%) film at room temperature; **b** a typical cross-sectional TEM image showing the columnar grain structure of the V$_{1-x}$W$_x$O$_2$/c-Al$_2$O$_3$ film; **c** and **d** atomic-resolution HAADF-STEM images showing different variants (rectangular and oval frames) of the pure M1 phase in the un-doped VO$_2$ sample, and the coexistence of M1 phase and R phase in the 3.44% W-doped sample, respectively.

[010]$_{M1}$//[010]$_R$//[0001]$_{Al2O3}$ relationship between V$_{1-x}$W$_x$O$_2$ and sapphire[38]. However, because other diffraction peaks are not visible, the XRD results obtained for the epitaxial V$_{1-x}$W$_x$O$_2$ film alone are not sufficient to provide evidence for the chemical substitution induced phase transformation. Therefore, transmission electron microscopy (TEM) was further employed to characterize the detailed microstructures in the V$_{1-x}$W$_x$O$_2$/c-Al$_2$O$_3$ samples.

**Microstructural characterization through TEM.** In order to further investigate the effect of chemical substitution in the composition-spread V$_{1-x}$W$_x$O$_2$/c-Al$_2$O$_3$ samples, TEM measurements were carried out on a number of cross-sectional samples extracted along the composition spread. A representative bright-field TEM image of the V$_{1-x}$W$_x$O$_2$/c-Al$_2$O$_3$ heterostructure is shown in Fig. 3b; the V$_{1-x}$W$_x$O$_2$ thin film has a near-uniform thickness of 150 nm and is composed of columnar grains. The analysis of selected area electron diffraction patterns (SAEDPs) evidences the [010]$_{M1}$//[010]$_R$//[0001]$_{Al2O3}$ epitaxial relationship for the V$_{1-x}$W$_x$O$_2$/c-Al$_2$O$_3$ heterostructure (Supplementary Fig. 1). Well-defined M1 phase variants were observed in the pure VO$_2$ strip without W-doping by aberration-corrected scanning transmission electron microscopy (STEM). Figure 3c shows an atomic-resolution high angle annular dark-field (HAADF) STEM image taken from the x = 0% strip, presenting a unique [101]$_{M1}$ atomic configuration and distinguishing it from the well-established M2 phase and high-temperature R phase, which are difficult to be differentiated from the M1 phase by SAEDPs (see Supplementary Fig. 2). The coexistence of the M1 and R phases was observed by HAADF-STEM imaging and SAEDPs in the W-doped sample strips (Fig. 3d and Supplementary Fig. 1). Figure 3d shows the mixture of the M1 and R

phases in a nano-sized region in the 3.44% W-doped sample. Therefore, although it is difficult to resolve the M1 phase and the R phase by XRD on c-Al$_2$O$_3$ (Fig. 3a), the HAADF-STEM results clearly demonstrate that W-doping leads to the formation of co-existing M1 and R phases[28].

It has been found that the intermediate M2 phase of VO$_2$ appears often in nanobeams[39] and sometimes in thin films[40,41]. The fact that the M2 phase is more prevalent in nanobeams is presumably owing to the stronger geometric effect in nano-beams[42]. Specifically, the M2 phase has been observed even in nanobeams where interfacial coupling between the nano-sized materials and the underneath substrates is negligible[43,44], suggesting that the intrinsic geometric effects owing to the high surface-to-volume ratios of these quasi-1-dimensional structures facilitate the formation of the intermediate phase through the MIT. Further, according to a number of previous studies[39,41,45,46], replacing V ions with a small amount (up to few %) of W is expected to reduce the transition temperature of the MIT without introducing the M2 phase through the structural transformation. Therefore, the absence of evidence of the M2 phase in our composition-spread films is consistent with previously V$_{1-x}$W$_x$O$_2$ work on thin films of V$_{1-x}$W$_x$O$_2$, where unlike in nanobeams, the M2 phase was not observed.

**Analysis of cofactor conditions based on crystal structures.** In lightly-doped V$_{1-x}$W$_x$O$_2$ films, the V/W ions in the high-temperature tetragonal R phase are aligned along the c axis of the crystal structure[46,47]. The structural phase transformation to the low-temperature M1 phase and, accordingly, the symmetry breaking results in the observing the V/W ions aligned into zig-zagging chains[48]. The inset of Fig. 4 illustrates the correspondence between two tetragonal unit cells in the R phase and one

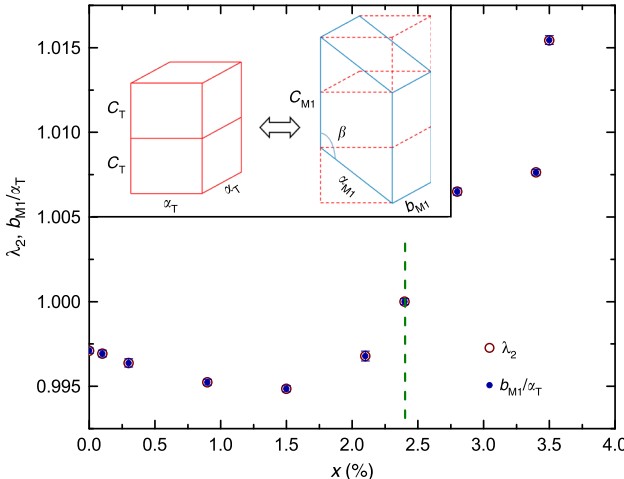

**Fig. 4 Concentration dependence of structural parameters.** The middle eigenvalue $\lambda_2$ of the stretch matrix (empty circles) and the $b_{M1}/a_T$ ratio (solid dots) are plotted as functions of the W concentration ($x$), respectively; the inset illustrates a schematic transformation from two tetragonal unit cells in the R phase to one monoclinic unit cell in the M1 phase, where $a_T$ and $c_T$ are the lattice parameters of the tetragonal unit cell, whereas $a_{M1}$, $b_{M1}$, $c_{M1}$, and $\beta$ are the lattice parameters of the monoclinic unit cell. A dashed vertical line indicates that $\lambda_2$ and $b_{M1}/a_T$ both reach a unit at $x \approx 2.4\%$.

monoclinic unit cell in the M1 phase through the structural transformation. It is important to note that, for illustrative purposes, the unit cells shown in the inset of Fig. 4 merely reflect the correspondence of the lattice parameters in the two-end phases without taking the lattice stretching effects into account. In fact, the actual lattice constants, which can be determined from the XRD results, are the key to the interfacial stress generated during the phase transformation.

To limit the interfacial stresses, which develop during the structural phase transformation, high compatibility between the two-end phases is required. According to the non-linear theory[5–8], when the cofactor conditions are satisfied, a highly compatible phase transformation becomes possible, resulting in the minimal thermal hysteresis width[4]. Because the transformation stretch matrix is solely determined by the structural relationship between the two-end phases, the XRD measurements and Rietveld refinement (an example of the refinement results is shown in Supplementary Fig. 3) were used to establish the crystal symmetries and lattice constants for different W concentrations across the composition-spread films. Supplementary Table 1 shows a list of the lattice parameters obtained at different W concentrations in the two-end phases.

A common geometric feature of the stretch tensors in Supplementary Table 1 is that the middle eigenvalue $\lambda_2$ is associated with the eigenvector aligned with the tetragonal $a$ axis. In this case, the kinematic compatibility conditions[8] ensure that when the lattice parameters are tuned to satisfy $\lambda_2 = 1$, a laminate of monoclinic compound twins is compatible with the tetragonal phase at arbitrary volume fraction of the twin pair. This is illustrated in Supplementary Fig. 4, using our measured lattice parameters at $x = 2.4\%$. Note the excellent matching of phases despite quite large distortions of ~6% strain. In particular, in the 1st and 6th schematic pictures of Supplementary Fig. 4 (i.e., Supplementary Fig. 4a, f), a single monoclinic variant matches the tetragonal lattice at a stress-free interface without any transition layer. Again, using our measured lattice parameters of $V_{1-x}W_xO_2$ at $x = 2.4\%$, we zoom in and plot the local structure of the interface in the region of the circle.

The complete analysis of the measured lattice parameters at different W concentrations provides the middle eigenvalue $\lambda_2$ of the transformation stretch matrix as a function of the W concentration (Fig. 4). We find that the $b_{M1}/a_T$ ratio (also shown in Fig. 4) is highly correlated with the middle eigenvalue $\lambda_2$ for all W concentrations, both approaching 1 simultaneously at a W concentration of ~2.4%. As shown in the inset of Fig. 4, the $b_{M1}$ lattice is not only normal to both $a_{M1}$ and $c_{M1}$ lattices in the M1 phase, but also corresponds to one of the $a_T$ lattices of the tetragonal unit cell of the R phase. The fact that the $b_{M1}/a_T$ ratio is always the middle eigenvalue $\lambda_2$ suggests that the stretch or compression through the structural phase transformation is mainly perpendicular to $b_{M1}$. Furthermore, the results also suggest that any stretch or compression along the direction of the $b_{M1}$ lattice would enhance the deformation in directions perpendicular to the $b_{M1}$ lattice. Therefore, only when the $b_{M1}/a_T$ ratio becomes 1 (i.e., the cofactor conditions are satisfied) at $x = 2.4\%$, the structural deformation in directions perpendicular to the $b_{M1}$ lattice is minimized, and phase transformation with minimal distortion becomes possible.

**Thermal hysteresis of electronic transport.** According to the non-linear theory[5–8], the fulfillment of the cofactor conditions is expected to minimize the interfacial energy involved during the phase transformation, and thus reduce the width of the hysteresis loop upon thermal cycling. Therefore, the thermal hysteresis width is expected to provide another measure for the compatibility between the two-end phases. Among various physical properties of $V_{1-x}W_xO_2$ that show a thermal hysteresis loop, the MIT provides a convenient route to quantify the transition temperature $T_C$ and the thermal hysteresis width $\Delta T$ (defined below).

Electrical measurements were performed on both epitaxial and polycrystalline films to investigate the W-substitution effect on the characteristics of MIT. Figure 5a, b show the hysteretic temperature-dependent sheet-resistance curves ($R_S$–$T$) of composition-spread thin films on $c$-$Al_2O_3$ and on Si, respectively. For both composition-spread films, the expected systematic reduction of the transition temperature with increase in the W concentration is seen. Specifically, as shown in Supplementary Fig. 5, the transition temperature shows a nearly linear decrease with the increase of W concentration for both films. The linear fits shown in Supplementary Fig. 5 further suggest that corresponding to each at. % increase in the W concentration, the transition temperatures in the epitaxial film and the polycrystalline film are reduced by 25 and 21 K, respectively, which agree well with the values reported previously[28,30].

We further extracted the $T_C$ and the thermal hysteresis width $\Delta T_C$ associated with the transition from the measured $R_S$–$T$ curves (Supplementary Fig. 6 shows an example). The $R_S$–$T$ curve obtained on a $VO_2$ (i.e., $x = 0\%$) strip fabricated on a $c$-$Al_2O_3$ substrate shows a change of nearly four orders of magnitude through the MIT. Two transition temperatures ($T_{Cup}$ and $T_{Cdn}$) were obtained by taking the temperatures at the dips in the first-order derivatives of the warming curve and the cooling curve, respectively[49]. Further, a phase transition temperature $T_C$ at ~342 K, consistent with that obtained from $VO_2$ single crystals[20], was determined as the average of the two transition temperatures, i.e., $T_C = (T_{Cup} - T_{Cdn})/2$. The hysteresis width $\Delta T_C$ is defined as the difference between the two transition temperatures, i.e., $\Delta T_C = T_{Cup} - T_{Cdn}$, which for this strip is 6.86 K.

It should be noted that the hysteresis width is known to be scan rate dependent for first-order transitions with latent heat in general. As shown in Supplementary Fig. 7, for a sample strip with a W concentration of 0.9%, the measured hysteresis width

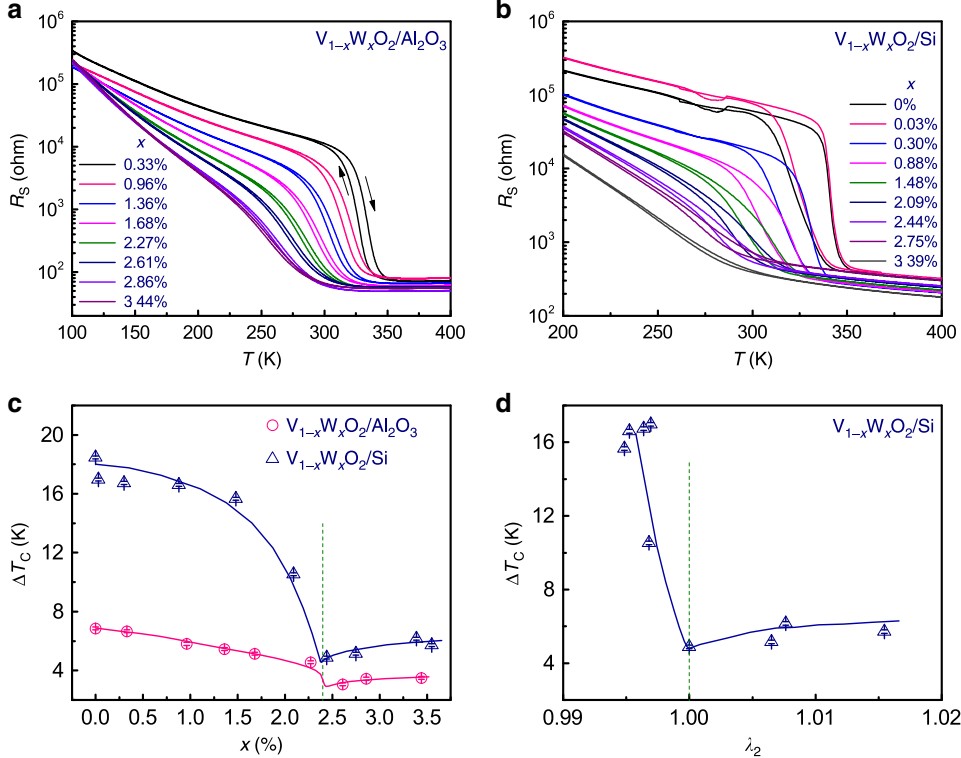

**Fig. 5 Thermal hysteresis of electronic transport. a, b** The temperature dependence of the resistance ($R_S$) measured from sample strips at different W concentrations in a $V_{1-x}W_xO_2$ film grown on $c$-$Al_2O_3$ and on Si, respectively. **c** The W content ($x$) dependence of hysteresis width ($\Delta T_C$) for both $V_{1-x}W_xO_2$/$Al_2O_3$ and $V_{1-x}W_xO_2$/Si films; **d** hysteresis width ($\Delta T_C$) as a function of $\lambda_2$ for $V_{1-x}W_xO_2$/Si. Solid curves in **c** and **d** are used for guidance; a dashed vertical line is placed at $x = 2.4\%$ in **c** to indicate the concentration where $\lambda_2$ and $b_{M1}/a_T$ both reach a unit; and a dashed vertical line is placed at $\lambda_2 = 1$ in **d**.

increases dramatically as the cooling and heating rate increases. In our resistance measurements, to limit the error caused by the latent heat, a relatively low scan rate of 1.0 K per minute was adopted. In addition, a universal scan rate also makes a meaningful comparison across the entire composition range of the sample possible.

In Fig. 5c, the measured $\Delta T_C$ was plotted as a function of W concentration for both films. The overall hysteresis width $\Delta T_C$ for the $V_{1-x}W_xO_2$/Si sample is clearly larger than that for the $V_{1-x}W_xO_2$/$c$-$Al_2O_3$ sample. The difference in the values of the hysteresis width $\Delta T_C$ may be attributed to the polycrystalline nature of the film grown on the Si substrate[47,50]. Despite the difference in the exact values of the measured $\Delta T_C$, both spread films share a common feature—there is a clear drop in $\Delta T_C$ at the W concentration of ≈2.4%, indicating the importance of satisfying the cofactor conditions in minimizing the thermal hysteresis width. Furthermore, as shown in Fig. 5d, the measured $\Delta T_C$ for the $V_{1-x}W_xO_2$/Si sample is also plotted as a function of $\lambda_2$. Clearly, as $\lambda_2$ becomes 1 (i.e., the cofactor conditions are satisfied), the thermal hysteresis width $\Delta T_C$ reaches the minimum value, thus confirming that fulfilling the cofactor conditions indeed leads to minimization of the thermal hysteresis width $\Delta T_C$. Therefore, as evident in the composition-spread $V_{1-x}W_xO_2$ films grown on different substrates, the non-linear theory of martensite is applicable not only to metallic systems but also to oxide systems.

It is important to note that our study demonstrates a straightforward route for meeting the cofactor condition using a single-element-substitution composition spread: tuning $\lambda_2$ to be (close to) 1; and the minimum thermal hysteresis width (as shown in Fig. 5c 3 and 5 K for the $V_{1-x}W_xO_2$/$c$-$Al_2O_3$ sample and the $V_{1-x}W_xO_2$/Si sample, respectively) obtained in this single-element-substitution composition spread is not necessarily a

universal minimum in doped $VO_2$ samples. In fact, Miyazaki et al. have obtained a hysteresis width as small as 0.6 K using both Cr and Nb to substitute V[51]. According to reported phase diagrams[39,46] of $VO_2$, W or Nb substitution leads to a reduction effect and also an increase in the lattice constant, and in comparison, using Cr to substitute V leads to an oxidation effect and also a decrease in the lattice constant. Therefore, it is reasonable to expect that by substituting V with Cr and Nb simultaneously, the volume change during the phase transformation is suppressed, resulting in a smaller minimum hysteresis width as compared with the case where only one substitution element is used.

## Discussion

In this study, high-quality composition-spread $V_{1-x}W_xO_2$ films were fabricated on $c$-$Al_2O_3$ and Si substrates using a high-throughput pulsed-laser deposition technique. XRD, TEM, and electronic transport measurements were performed to systematically investigate the W-substitution effect on the structural phase transformation and the MIT. Based on the lattice parameters determined from the XRD measurements at different temperatures, we found that the cofactor conditions based on the geometrically non-linear theory of martensite are satisfied at a W concentration of 2.4%. The measurements of the MIT in the spreads indicate that the thermal hysteresis width indeed reaches the minimum value for the samples with W concentration near 2.4%.

The first-order transition in $VO_2$ is a complex process involving change in the electronic structure as well as transformation of the crystal structure accompanied by a large latent heat. The subtle interplay between various factors is reflected in the sensitivity of the transformation temperature and its hysteresis to

small chemical modifications and geometrical effects (e.g., films versus nanobeams) including strain. The fact that lattice compatibility has influence on the hysteresis as observed here underscores the basic role crystal structure plays in determining the physical properties of materials in general.

The success of applying the non-linear theory to identify the conditions for the ultra-compatible MIT in a functional oxide system suggests that the theory is highly valuable in guiding the optimization of transforming materials. Moreover, the correlation between the MIT and the crystal structure demonstrated in our study also suggests that the structural phase transformation has a central role in the observed MIT in the $V_{1-x}W_xO_2$ system.

## Methods

**Combinatorial deposition and composition characterization**. The continuous composition-spread films of $V_{1-x}W_xO_2$ ($0 \leq x < 4.0\%$) used in this study were fabricated in a combinatorial pulsed-laser deposition chamber. During the deposition, the substrate temperature was kept at ~500 °C, an oxygen environment with a pressure of ~0.4 Pa was applied, and laser pulses with a frequency of 5 Hz and an energy of 18 mJ were used for material ablation. Moreover, in order to minimize the substrate-induced strain, the composition-spread films used in this study all had a thickness of ≈150 nm. Wavelength dispersive X-ray spectroscopy was performed to characterize the W concentration in different positions of the composition-spread $V_{1-x}W_xO_2$ films.

**X-ray diffraction measurements**. The XRD measurements were carried out in a Bruker D8 Discover system. The system was equipped with an area detector and a stage that allows the sample's translation in the direction of a composition gradient for automated data collection. With in situ temperature control of the sample using either a heater or a liquid-nitrogen cold bath attached to the sample stage, $\theta$–$2\theta$ X-ray spectra were collected at several temperatures from 255 to 358 K.

**Sample preparation for cross-sectional TEM/STEM**. A FEI Nova NanoLab 600 dual-beam scanning electron microscopy and focused ion-beam system was employed to prepare the cross-sectional TEM/STEM samples. Regions of 20 μm in length in the $V_{1-x}W_xO_2$ ($x = 0$, 2.61%, 3.44%) strips were chosen for lamellar TEM/STEM samples preparation. Electron-beam induced deposition of 1 μm thick Carbon was initially deposited on top of the film to protect the sample surface, then followed by 2 μm ion-beam induced Pt deposition. To reduce Ga-ions damage, in the final stage of FIB preparation, the TEM/STEM sample was thinned with 2 kV Ga-ions using a low beam current of 29 pA and a small incident angle of 3°.

**TEM/STEM characterization**. A FEI Titan 80–300 TEM/STEM equipped with a probe spherical-aberration corrector was employed to conduct selected area electron diffraction patterns (SAEDPs), diffraction-contrast imaging (DFI) and atomic-resolution HAADF STEM imaging analyses. HAADF-STEM images were acquired with an operating voltage of 300 kV, probe convergence semi-angle of 14 mrad and collection angle of (70–400) mrad.

**Electronic transport measurements**. The temperature dependence of the electrical resistance was measured for each sample strip in the $V_{1-x}W_xO_2$ spread films by utilizing a four-probe geometry using a physical property measurement system made by Quantum Design Inc. To improve the measurement accuracy, temperature scans were all carried out with a sweeping rate of 1.0 K per minute and a measurement step of 0.2 K.

## Disclaimer

Certain commercial equipment, instruments, or materials are identified in this paper in order to specify the experimental procedure adequately. Such identification is not intended to imply recommendation or endorsement by the National Institute of Standards and Technology, nor is it intended to imply that the materials or equipment identified are necessarily the best available for the purpose.

## Data availability

All data generated or analyzed during this study are included in this published article (and its supplementary information files).

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

## Acknowledgements

This work was supported by ONR MURI N000141310635, ONR MURI N000141712661, and National Institute of Standards and Technology (NIST) Cooperative Agreement 70NANB17H301 at UMD. X.C. thanks the HK Research Grants Council for financial support under Grants 26200316 and 16207017. R.D.J. was supported by NSF (DMREF-1629026), ONR (N00014-18-1-2766), MURI (FA9550-18-1-0095) and a Vannevar Bush Fellowship. He also benefited from the support of Medtronic Corp, the Institute on the Environment (RDF fund), and the Norwegian Centennial Chair Program. X.C. and R.D.J. thank the Isaac Newton Institute for Mathematical Sciences for support and hospitality during the program "The mathematical design of new materials" (EPSRC EP/R014604/1) when work on this paper was undertaken. H.R.Z. acknowledges support from the U.S. Department of Commerce, NIST under the financial assistance awards 70NANB17H249 and 70NANB19H138. A.V.D. acknowledges the support of Material Genome Initiative funding allocated to NIST.

## Author contributions

Y.G.L., S.L., X.H.Z., and I.T. conceived and designed the experiments; Y.G.L. fabricated samples for the experiments; S.L., Y.G.L and H.S.Y. measured electronic resistivity; Y.G.L., Y.J.L., and P.Y.Z. carried out XRD measurements and crystal structure analysis; H.R.Z. conducted TEM samples FIB preparation and TEM/STEM experiments. H.R.Z., L.A.B., and A.V.D performed TEM/STEM microstructure analysis. X.C. and R.D.J. provided numerical simulation; Y.G.L., S.L., and X.H.Z. summarized the data and prepared figures; Y.G.L., S.L. X.H.Z., and I.T. wrote the manuscript; and all authors participated in discussion of the results and preparation of the manuscript.

## Competing interests

## Additional information

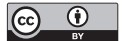

