## [Peer Review File · Nature Communications]

Reviewers' comments:

Reviewer #1 (Remarks to the Author):

The authors studied $WxV_{1-x}O_2$ strips grown on Si and c-Al₂O₃ substrates, and observed that the hysteresis of MIT curves can be tuned by tungsten doping. By applying the non-linear theory to $WxV_{1-x}O_2$, the authors explained the reason of hysteresis variation, and provided a possible way to minimize the hysteresis and optimize the phase transformation performance. The results are simple yet elegant and clean. A few questions need to be addressed:

1. The authors prepared lateral graded doped $WxV_{1-x}O_2$ thin films, then patterned strips with different x . How were x values of these strips determined? BTW, the text said "...patterned into multiple strips along the composition gradient direction...", while it should be corrected as "...patterned into multiple strips perpendicular to the composition gradient direction...".
2. The width of the strips is 0.2mm, which is not very small. Can the lateral x difference be neglected at this scale? The resultant error bars in x in their Fig.4 and Fig.5 should then be equal to the difference between x of each data points, i.e., +/- 0.25%. Please add the error bars.
3. The $WxV_{1-x}O_2$ strips are grown on Si/ c-Al₂O₃, what is the role of the substrates? If the measurements are performed on single-crystalline, free standing $WxV_{1-x}O_2$, will they give the same result? If not, need a thorough discussion on and outlook of using substrate for this purpose. This is what is useful for readers.
4. Which sample the data in Table. S1 corresponds to, $WxV_{1-x}O_2/Si$ or $WxV_{1-x}O_2/c-Al_2O_3$?

Reviewer #2 (Remarks to the Author):

The idea of relating lattice compatibility tensors, specifically the middle eigenvalue and co-factor conditions across Martensitic transitions, to obtain a strain-free interface between the martensite and austenite, and to thereby suppress hysteresis has been demonstrated elegantly in previous work by these authors in the case of two shape memory alloy systems. These ideas of lattice compatibility are sought to be extended here to electron correlated oxides. While the article demonstrates phenomenological validation of the core ideas of the model to some extent, claims of universality of the this approach need must be tempered by the relatively modest suppression of the hysteresis (a 4K hysteresis remains even under the best conditions, which is orders of magnitude higher than demonstrated for the shape memory alloys or even previously achieve with VO₂), the almost complete degradation of the magnitude of the metal—insulator transition for the optimal samples, absence of any demonstration beyond a single cycle, and the extremely slow scan rates utilized in this work. Below, I've provided a pointwise list of issues that ought to be addressed before I am able to recommend this work for publication in Nature Communications:

(1) The entire premise of this work, as laid out in Figure S1, is that there are several monoclinic variants corresponding to the low-temperature polymorph and a rutile phase corresponding to the high temperature phase of VO₂. What this work seems to neglect is the idea, now quite well established, of several energetically proximate polymorphs in VO₂, e.g., the M2 and triclinic phases that are known to be stabilized under conditions of strain and are nucleated upon approaching the phase transition. The idea of a "triple point" first established in work by Cobden also makes VO₂ a rather distinctive system that is quite different both in terms of its structural phase diagram and strain coupling as compared to the shape memory alloys with which the authors make their comparisons. I

provide here a partial list of articles: (a) Strelcov, E.; Ievlev, A.; Belianinov, A.; Tselev, A.; Kolmakov, A.; Kalinin, S. V., Local coexistence of VO₂ phases revealed by deep data analysis. *Scientific Reports* 2016, 6, 29216; (b) Braham, E. J.; Sellers, D.; Emmons, E.; Villarreal, R.; Asayesh-Ardakani, H.; Fleeer, N. A.; Farley, K. E.; Shahbazian-Yassar, R.; Arròyave, R.; Shamberger, P. J.; Banerjee, S., Modulating the Hysteresis of an Electronic Transition: Launching Alternative Transformation Pathways in the Metal–Insulator Transition of Vanadium(IV) Oxide. *Chemistry of Materials* 2018, 30 (1), 214–224; (c) Shi, Y.; Xue, F.; Chen, L.-Q., Ginzburg-Landau theory of metal-insulator transition in VO₂: The electronic degrees of freedom. *EPL (Europhysics Letters)* 2017, 120 (4), 46003; (d) Tselev, A.; Luk'yanchuk, I. A.; Ivanov, I. N.; Budai, J. D.; Tischler, J. Z.; Strelcov, E.; Kolmakov, A.; Kalinin, S. V., Symmetry Relationship and Strain-Induced Transitions between Insulating M1 and M2 and Metallic R phases of Vanadium Dioxide. *Nano Letters* 2010, 10 (11), 4409–4416.

Indeed, the stabilization of the M2 phase of VO₂ upon W incorporation has been extensively documented and directly observed in studies of W-doped (or for that matter, undoped) VO₂ nanobeams. If lattice strain can drive a transition to entirely different polymorphs (M2 and T) and not just a twin variant (the ORNL group has shown using extensive symmetry analysis that the M2 polymorph is not in fact a twin variant but a distinctive polymorph), the simple biphasic model of Figure S1 does not hold. Indeed, I suspect that if the authors were to undertake a detailed examination of the systems under consideration here, they would indeed find these polymorphs and it is the existence of these polymorphs that results in the substantial remnant hysteresis.

(2) The X-ray diffraction data shown in the manuscript corresponds to a narrow range of 2-theta values and thus it is impossible to rule out the presence of M2 and triclinic phases within the films, which are especially expected to be observed when approaching the transition temperature. Given the relative low intensities of reflections, Raman microprobe analysis performed as a function of temperature might perhaps be the best way to examine whether the simplified view presented here of M1/T transformations, which ignores extensive work on M2 nucleation in VO₂ thin films (particularly epitaxial films), is indeed credible.

(3) I find it puzzling that the authors are reporting results of Rietveld refinements without actually showing their data and the quality of the fits. Based on the limited diffraction data that has been presented, I expect that large error bars are warranted for Figure 4.

(4) The remnant hysteresis observed under the optimal conditions is >4K, which is substantially higher than results observed for the shape memory alloys and indeed this is a matter that deserve further comment. If the “happy accident” makes VO₂ such a perfect material, why then cannot the hysteresis be reduced further. Indeed, the results here are entirely not a record even for VO₂: for example, the following work has noted the almost complete elimination of hysteresis upon co-doping: Miyazaki, K.; Shibuya, K.; Suzuki, M.; Sakai, K.; Fujita, J.-I.; Sawa, A., Chromium–niobium co-doped vanadium dioxide films: Large temperature coefficient of resistance and practically no thermal hysteresis of the metal–insulator transition. *AIP Advances* 2016, 6 (5), 055012. I suspect the strain induced nucleation of other phases has a lot to do with the remnant hysteresis and ultimately represents a substantial limitation of this approach.

(5) Two critical parameters need further elucidation: (a) what happens at faster scan rates and (b) during subsequent cycles? The hysteresis in W-doped VO₂ is known to be scan rate dependent, reflective in some measure, the available nucleation sites. Further work has shown the kinetic asymmetry between the forward and reverse transitions derived from their distinctive origins (Fan, W.; Cao, J.; Seidel, J.; Gu, Y.; Yim, J. W.; Barrett, C.; Yu, K. M.; Ji, J.; Ramesh, R.; Chen, L. Q.; Wu, J. Large Kinetic Asymmetry in the Metal-Insulator Transition Nucleated at Localized and Extended Defects. *Phys. Rev. B - Condens. Matter Mater. Phys.* 2011, 83 (23), 235102-7.). Lattice strain derived from epitaxial matching strongly alters the formation of twin boundaries and as a result modifies the nucleation probabilities of the M2 and T polymorphs. I expect this explains the differences the authors observe between the films grown on Si and Al₂O₃.

(6) In Figure 5b, there seems to be no formal metal–insulator transition. The transport data indicate

a semiconductor-semiconductor transition. Can the authors explain the strong suppression of the magnitude of the transition and its abruptness (indicating a protracted phase coexistence regime) in these samples (which is not in fact observed for high-quality W-doped VO₂ films and nanobeams). How repeatable are these measurements? Past calorimetry measurements reported by the authors have gone out to thousands of cycles but only singular sweeps are shown here.

(7) In a similar context of the above point, many measurements of VO₂ thin films and nanobeams indicate a distinctive step-like alteration of transport corresponding to switching of individual domains—the extended and broad coexistence regimes observed here are likely due to a complex domain microstructure (can this be imaged?) and the stabilization of various polymorphs. If such texturation is indeed observed and given the substantial degradation of the magnitude of the phase transition, this would negate some of the intended benefits of hysteresis reduction.

(8) Have the authors attempted to image domain evolution within these samples? Considerable work on phase contrast microscopy is now available and given the broad transitions, I expect that the diminished hysteresis has much to do with the facilitation of percolative mesoscale textures through this system.

(9) As a semantic but nonetheless important point, VO₂ is not a representative electron correlated material given the substantial role of phonons in underpinning the MIT (see for example, Budai, J.D., Hong, J., Manley, M.E., Specht, E.D., Li, C.W., Tischler, J.Z., Abernathy, D.L., Said, A.H., Leu, B.M., Boatner, L.A., McQueeney, R.J., and Delaire, O. (2014) Metallization of vanadium dioxide driven by large phonon entropy. *Nature* 515, 535–539). As such, extrapolating from VO₂ to Mott systems wherein electron correlation effects are of paramount importance is entirely unjustified and not supported by any of the data presented here.

(10) Can the authors provide example of representative WDS data and discuss the accuracy of this quantitation approach. I note that there are no horizontal error bars in Figure 4.

Reviewer #3 (Remarks to the Author):

The authors applied the non-linear theory of martensite to a strongly correlated oxide system V(1-x)WxO₂, showing that the hysteresis of the MIT is minimized when the cofactor conditions are satisfied. The minimized hysteresis may feature high reversibility of the material, which is desired in applications.

This work provides a systematic way to reduce the hysteresis of the MIT in W-doped VO₂ and improves our understanding of the hysteresis of the MIT. It will be of interest to the MIT research community. Below are my comments on this work.

1. The local structure around the W dopant is a tetragonal-like structure [Scientific Reports 2, 466 (2012)], which may serve as nucleation sites at phase transitions. This will have an effect on the hysteresis: more nucleation sites are likely to lead to smaller hysteresis. Compared to the lattice compatibility between phases, what is the role of this effect in regulating the hysteresis?

2. Intuitively, the more deviation from 1 for $bM1/aT$, the larger lattice incompatibility between phases, and thus the larger hysteresis. This is however not the case according to Fig. 4 and Fig. 5. Can the authors discuss this point? Maybe this is related to the first comment, that is, besides the lattice compatibility aspect, there is another aspect (nucleation sites) that can influence the hysteresis.

3. Figure 5(c) shows that the hysteresis of the polycrystalline V(1-x)WxO₂/Si is larger than that of the epitaxial V(1-x)WxO₂/Al₂O₃. But in general better crystals should exhibit larger hysteresis due to fewer nucleation sites. Can the authors give a more detailed explanation for the measured result?

Reviewer #1 (Remarks to the Author):

The authors studied $WxV1-xO2$ strips grown on Si and c-Al₂O₃ substrates, and observed that the hysteresis of MIT curves can be tuned by tungsten doping. By applying the non-linear theory to $WxV1-xO2$, the authors explained the reason of hysteresis variation, and provided a possible way to minimize the hysteresis and optimize the phase transformation performance. The results are simple yet elegant and clean. A few questions need to be addressed:

1. The authors prepared lateral graded doped $WxV1-xO2$ thin films, then patterned strips with different x . How were x values of these strips determined? BTW, the text said "...patterned into multiple strips along the composition gradient direction...", while it should be corrected as "...patterned into multiple strips perpendicular to the composition gradient direction...".

Yes - this was a confusing description. The length of each strip is along the direction perpendicular to the composition gradient direction. We have corrected the sentence accordingly.

2. The width of the strips is 0.2mm, which is not very small. Can the lateral x difference be neglected at this scale? The resultant error bars in x in their Fig.4 and Fig.5 should then be equal to the difference between x of each data points, i.e., +/- 0.25%. Please add the error bars.

Along the long direction of each sample, the composition is continuously varied from $x = 0 \%$ to $x \approx 4.0 \%$ within a 7 mm range at the center of the substrate. Therefore, given a width of 0.2 mm, each strip is expected to have a variation of $\sim 0.2/7 \times 4.0 \%$ in the W concentration across the entire width of the strip: that is, $|\Delta x|$ is expected to be less than 0.1 %. In addition, we should also take the measurement error of the WDS into account. We took the average of multiple measured values as the concentration for each strip, and the spreads (standard deviation) within the measured values were found to be consistent with the above estimate. We plot this in FIG. R1 below. In the first paragraph on page 6 of the revised manuscript, we now indicate that "... according to the change in the composition across the entire substrate and the width of each strip, the variation of the W concentration within each strip is estimated to be $\pm 0.1\%$ ".

FIG. R1. W concentration (x) as a function of the distance from the sample edge of the pure VO_2 side for a $\text{V}_{1-x}\text{W}_x\text{O}_2$ sample made on Si. The concentration value at each position is an average of multiple measurements within the sample strip, and the uncertainty corresponds to the standard deviation of the measured values within each strip.

3. The $\text{W}_x\text{V}_{1-x}\text{O}_2$ strips are grown on Si/ $c\text{-Al}_2\text{O}_3$, what is the role of the substrates? If the measurements are performed on single-crystalline, free standing $\text{W}_x\text{V}_{1-x}\text{O}_2$, will they give the same result? If not, need a thorough discussion on and outlook of using substrate for this purpose. This is what is useful for readers.

We performed X-ray and electronic transport measurements on both polycrystalline samples fabricated on Si and epitaxial samples fabricated on $c\text{-Al}_2\text{O}_3$. We also performed TEM studies on epitaxial samples fabricated on $c\text{-Al}_2\text{O}_3$. Epitaxial films generally represent “higher” quality films (compared to polycrystalline films) as they can be single-crystal-like in crystallinity and properties. Epitaxial films are, however, by definition, of single orientation and lacking diffraction peaks from other orientations (in standard Bragg-Brentano geometry.) Because an essential part of the present work is accurate

determination of the lattice constants (necessary to apply the non-linear theory), we have elected to carry out detailed XRD diffraction analysis and accompanying refinement on polycrystalline samples (i.e., the data obtained from films fabricated on Si substrates). The out-of-plane lattice constants we measure on epitaxial films on $c\text{-Al}_2\text{O}_3$ are consistent with those obtained on polycrystalline films.

4. Which sample the data in Table. S1 corresponds to, $\text{W}_x\text{V}_{1-x}\text{O}_2/\text{Si}$ or $\text{W}_x\text{V}_{1-x}\text{O}_2/c\text{-Al}_2\text{O}_3$?

The analysis shown in Table S1 corresponds to a $\text{V}_{1-x}\text{W}_x\text{O}_2$ sample fabricated on Si. We regret that this was not clearly indicated in the earlier version of the manuscript. In the revised manuscript, we now indicate that the structural analyses shown in Table 1 were obtained on a $\text{V}_{1-x}\text{W}_x\text{O}_2/\text{Si}$ sample.

Reviewer #2 (Remarks to the Author):

The idea of relating lattice compatibility tensors, specifically the middle eigenvalue and co-factor conditions across Martensitic transitions, to obtain a strain-free interface between the martensite and austenite, and to thereby suppress hysteresis has been demonstrated elegantly in previous work by these authors in the case of two shape memory alloy systems. These ideas of lattice compatibility are sought to be extended here to electron correlated oxides. While the article demonstrates phenomenological validation of the core ideas of the model to some extent, claims of universality of the this approach need must be tempered by the relatively modest suppression of the hysteresis (a 4K hysteresis remains even under the best conditions, which is orders of magnitude higher than demonstrated for the shape memory alloys or even previously achieve with VO₂), the almost complete degradation of the magnitude of the metal–insulator transition for the optimal samples, absence of any demonstration beyond a single cycle, and the extremely slow scan rates utilized in this work. Below, I've provided a pointwise list of issues that ought to be addressed before I am able to recommend this work for publication in Nature Communications:

*(1) The entire premise of this work, as laid out in Figure S1, is that there are several monoclinic variants corresponding to the low-temperature polymorph and a rutile phase corresponding to the high temperature phase of VO₂. What this work seems to neglect is the idea, now quite well established, of several energetically proximate polymorphs in VO₂, e.g., the M2 and triclinic phases that are known to be stabilized under conditions of strain and are nucleated upon approaching the phase transition. The idea of a “triple point” first established in work by Cobden also makes VO₂ a rather distinctive system that is quite different both in terms of its structural phase diagram and strain coupling as compared to the shape memory alloys with which the authors make their comparisons. I provide here a partial list of articles: (a) Strelcov, E.; Ievlev, A.; Belianinov, A.; Tselev, A.; Kolmakov, A.; Kalinin, S. V., Local coexistence of VO₂ phases revealed by deep data analysis. *Scientific Reports* 2016, 6, 29216; (b) Braham, E. J.; Sellers, D.; Emmons, E.; Villarreal, R.; Asayesh-Ardakani, H.; Fler, N. A.; Farley, K. E.; Shahbazian-Yassar, R.; Arròyave, R.; Shamberger, P. J.; Banerjee, S., Modulating the Hysteresis of an Electronic Transition: Launching Alternative Transformation Pathways in the Metal–Insulator Transition of Vanadium(IV) Oxide. *Chemistry of Materials* 2018, 30 (1), 214-224; (c) Shi, Y.; Xue, F.; Chen, L.-Q., Ginzburg-Landau theory of metal-insulator transition in VO₂: The electronic degrees of freedom. *EPL (Europhysics Letters)* 2017, 120 (4), 46003; (d) Tselev, A.; Luk'yanchuk, I. A.; Ivanov, I. N.; Budai, J. D.; Tischler, J. Z.; Strelcov, E.; Kolmakov, A.; Kalinin, S. V., Symmetry Relationship and Strain-Induced Transitions between Insulating M1 and M2 and Metallic R phases of Vanadium Dioxide. *Nano Letters**

2010, 10 (11), 4409-4416.

Indeed, the stabilization of the M2 phase of VO₂ upon W incorporation has been extensively documented and directly observed in studies of W-doped (or for that matter, undoped) VO₂ nanobeams. If lattice strain can drive a transition to entirely different polymorphs (M2 and T) and not just a twin variant (the ORNL group has shown using extensive symmetry analysis that the M2 polymorph is not in fact a twin variant but a distinctive polymorph), the simple biphasic model of Figure S1 does not hold. Indeed, I suspect that if the authors were to undertake a detailed examination of the systems under consideration here, they would indeed find these polymorphs and it is the existence of these polymorphs that results in the substantial remnant hysteresis.

As the reviewer points out, the presence of ‘proximate polymorphs’, such as the M2 and the triclinic phases, has been observed in VO₂ samples when internal or external stress is introduced. These intermediate phases are more prevalent in nanobeams [Tselev, A. *et al.*, Nano Letters **10**, 4409 (2010)], although occasionally also observed in thin films [Ji, Y. *et al.*, Scientific Reports **4**, 4854 (2014); Tan, X. *et al.*, Scientific Reports **2**, 466 (2012)]. In particular, according to various reports [Ji, Y. *et al.*, Scientific Reports **4**, 4854 (2014), Okimura, K. *et al.*, J. Appl. Phys. **107**, 063503 (2010)], it is often the case that the M2 phase or the triclinic phase is not found in as-grown epitaxial thin films at room temperature, but can be introduced as an intermediate phase during thermal cycling between the R phase and the M1 phase.

To look for presence of such phases in our films more carefully, we have gone back and performed extensive transmission electron microscopy (TEM) study on our thin-film samples at room temperature. The conclusion is that there is no clear evidence of the presence of the M2 or the triclinic phase in our thin films (see FIG. S2 and FIG. S3 in Supplementary Information). This result is consistent with the reported absence of the M2 phase in as-grown V_{1-x}W_xO₂ films epitaxially fabricated on Al₂O₃ substrates, where moderate interfacial stress (e.g. external stress) is expected [Ji, Y. *et al.*, Scientific Reports **4**, 4854 (2014)].

For internal stress introduced through chemical substitution, previous studies [see Pouget, J. P. and Launois, H., J. Phys. Colloques **37**, C4-49 (1976); and also see FIG. 1 in Strelcov, E. *et al.*, Nano Letters **12**, 6198 (2012)] have discussed the correlation between chemical substitution and uniaxial stress in terms of influencing the temperature-composition phase diagram. Specifically, as shown by Strelcov *et al.*, replacing V with a small amount (up to few %) of W is expected to reduce the transition temperature of the MIT (as we observe in this work) *without* introducing the M2 phase through a structural transformation. This is different, for instance, from the case of Cr substitution of V, where the M2 phase emerges upon substitution, while the transition temperature increases with increased substitution level.

Therefore, the absence of the M2 phase in our W-substituted composition spread films is entirely consistent with the previous work, and not surprising.

Given that the M2 phase has been reported to emerge as an intermediate phase in stressed pure VO₂ systems during thermal cycling of the phase transformation, it is natural to ask whether the M2 phase (and/or the triclinic phase) appears during the thermal cycling of our films of different compositions. To this end, we also performed more systematic room-temperature TEM measurements on composition regions where the hysteresis loop (FIG. 5) covers the measurement temperature (that is, for $x \approx 2-3\%$). As shown in FIG. S2, our results show no evidence of the presence of the M2 phase or the triclinic phase. Their absence in our thin films is again consistent with the phase diagram mentioned above [Pouget, J. P. and Launois, H., J. Phys. Colloques **37**, C4-49 (1976); and Strelcov, E. *et al.*, Nano Letters **12**, 6198 (2012)]: that is, W substitution does not introduce the intermediate phases in the V_{1-x}W_xO₂ system. Moreover, after over 10 thermal cycling of the sample, the room-temperature TEM shows no evidence of the presence of the intermediate M2 in the unsubstituted VO₂ composition either.

Thus, based on our extensive revisited TEM work where we went out of our way to search for the M2 phase in various compositions and in some instances, after thermal cycling, we did not find any. Given the large number of locations we carefully studied across various samples and the absence of the M2 phase at these locations, we place the upper limit of the volume fraction of (possible) trace amount of this intermediate M2 phase at one percent. Again, this is entirely consistent with previous work on *thin films* of V_{1-x}W_xO₂, where unlike in nanobeams, the M2 phase was not observed.

In the revised manuscript, a detailed discussion on the possible presence of the M2 phase (and the lack thereof) has been added, and new supplementary figures (FIG. S2 and FIG. S3 in the revised manuscript) have been added. FIG. 3c and FIG. 3d have also been revised. In particular, on Page 8 of the revised manuscript, the TEM section now reads as:

“Microstructural characterization through TEM. In order to further investigate the effect of chemical-substitution in the composition-spread V_{1-x}W_xO₂/c-Al₂O₃ samples, transmission electron microscopy (TEM) measurements were carried out on a number of cross-sectional samples extracted along the compositional gradient. A representative bright-field TEM image of the V_{1-x}W_xO₂/c-Al₂O₃ heterostructure is shown in FIG. 3b; the V_{1-x}W_xO₂ thin film has a near-uniform thickness of 150 nm and is composed of columnar grains. The analysis of selected area electron diffraction patterns (SAEDPs) indicates the [010]_{M1}//[010]_R//[0001]_{Al2O3} epitaxial relationship of the V_{1-x}W_xO₂/c-Al₂O₃ heterostructure (FIG. S2). Well-defined M1 phase variants were observed in the pure VO₂ (with no W doping) strip by

aberration-corrected scanning transmission electron microscopy (STEM). FIG. 3c shows an atomic-resolution high angle annular dark-field (HAADF) STEM image taken from the $x = 0$ % strip, presenting a unique $[101]_{M1}$ atomic configuration and distinguishing it from the well-established M2 phase and the high temperature R-phase which are difficult to differentiate from the M1 phase by SAEDPs (see FIG. S3). The coexistence of the M1 and R phases was observed by HAADF-STEM imaging and SAEDPs in the W-doped sample strips (FIG. 3d and FIG. S2). FIG. 3d shows the mixture of the M1 and R phases in a nano-size region in the 3.44 % W-doped sample. Therefore, although it is difficult to resolve the M1 phase and the R phase by XRD on $c\text{-Al}_2\text{O}_3$ (FIG. 3a), the HAADF-STEM results clearly demonstrate that W-doping leads to the formation of co-existing M1 and R phases²⁷.

It has been found that the intermediate M2 phase of VO_2 appears often in nanobeams³⁸ and sometimes in thin films^{39,40}. The fact that the M2 phase is more prevalent in nanobeams is presumably due to the stronger surface stress effect in nanobeams⁴¹. Further, according to a number of previous studies^{38, 40, 42, 43}, replacing V ions with a small amount (up to few %) of W is expected to reduce the transition temperature of the MIT without introducing the M2 phase through the structural transformation. Therefore, the absence of evidence of the M2 phase in our composition-spread films is consistent with previously $\text{V}_{1-x}\text{W}_x\text{O}_2$ work on *thin films* of $\text{V}_{1-x}\text{W}_x\text{O}_2$, where unlike in nanobeams, the M2 phase was not observed.”

The revised FIG. 3 together with the new FIG. S2 and FIG. S3 are duplicated below with corresponding captions.

FIG. 3 XRD and TEM/STEM results obtained from sample strips grown on a c - Al_2O_3 (0001) substrate. (a) XRD θ - 2θ patterns obtained from different sample strips of a $V_{1-x}W_xO_2$ ($0 \leq x < 4.0\%$) film at room temperature; (b) a typical cross-sectional TEM image showing the columnar grain structure of the $V_{1-x}W_xO_2/c$ - Al_2O_3 film; (c) and (d) atomic-resolution HAADF-STEM images showing the pure M1 phase in the un-doped VO_2 sample, and the co-existence of M1 phase and R-phase in the 3.44% W-doped sample, respectively.

FIG. S2. Typical SAEDPs taken from the epitaxial film of 2.61 % and 3.44 % W-doped $V_{1-x}W_xO_2/c$ - Al_2O_3 heterostructure. Al_2O_3 substrate was orientated to $[10-10]$ zone-axis (a, c), and $[2-1-10]$ zone-axis (b, d), respectively. The arrows shown in (a) and (c) indicate that the splitting reflections can be indexed by multiple options belonging to the variants of M1 phase or R phase. The arrows in (b) and (d) indicate that the splitting reflections can be well indexed with the M1 phase and R phase.

FIG. S3. Simulated electron diffraction patterns and the corresponding atomic projection of V atoms along the respective zone-axis. The patterns correspond to the possible variants of M1, M2 and R phases in the epitaxial $V_{1-x}W_xO_2/Al_2O_3$ film when Al_2O_3 is orientated to $[10-10]$ zone-axis (a), and $[2-1-10]$ zone-axis (b), respectively.

(2) *The X-ray diffraction data shown in the manuscript corresponds to a narrow range of 2-theta values and thus it is impossible to rule out the presence of M2 and triclinic phases within the films, which are especially expected to be observed when approaching the transition temperature. Given the relative low intensities of reflections, Raman microprobe analysis performed as a function of temperature might perhaps be the best way to examine whether the simplified view presented here of M1/T transformations, which ignores extensive work on M2 nucleation in VO₂ thin films (particularly epitaxial films), is indeed credible.*

As in our reply to the previous comment, we have carefully evaluated the possible effect of the M2 phase in the phase transformation. Our main experimental approach for the investigation is TEM, which is a reliable and well-accepted tool for the study of micro/nano sized phases in materials [Braham, E. J. *et al.*, Chem. Mater. **30**, 214 (2018)].

As discussed extensively above, our TEM investigation and search of the M2 and the triclinic phases strongly suggests that these intermediate phases are likely not playing a role in the observed structural phase transformation in our V_{1-x}W_xO₂ films. Again, this is consistent with previous reports by other groups on V_{1-x}W_xO₂ films [see Tan *et al.*, Scientific Reports **2**, 466 (2012); Rajeswarana and Umarji, AIP Advances **6**, 035215 (2016); Pouget, J. P. and Launois, H., J. Phys. Colloques **37**, C4-49 (1976); and Strelcov, E. *et al.*, Nano Letters **12**, 6198 (2012)].

(3) *I find it puzzling that the authors are reporting results of Rietveld refinements without actually showing their data and the quality of the fits. Based on the limited diffraction data that has been presented, I expect that large error bars are warranted for Figure 4.*

We carried out Rietveld refinements for the XRD patterns to obtain the lattice parameters. For the refinements, we used the XRD diffraction pattern of the polycrystalline VO₂ thin films fabricated on (SiO₂) Si substrates to ensure the observation of multiple diffraction peaks within the 2θ range of 25 – 60° in order to accurately calculate the lattice parameters. In FIG. R2, we show a representative XRD diffraction pattern of V_{1-x}W_xO₂ (x = 0.009) thin films and the Rietveld refinement result. Since $\lambda_2 = b_{M1}/a_T$ is only a function of the lattice parameters of M1 and rutile phases, there is no significant error as long as peak positions are captured accurately in the refinement. Although the lattice parameters can be obtained using Bragg's law and *d*-spacings determined by a crystal structure, we further carried out Rietveld

refinements as confirmation. In the revised manuscript, we now include FIG. R2 as FIG. S4 in the Supplementary Information.

FIG. R2. XRD diffraction patterns of $V_{1-x}W_xO_2$ ($x=0.009$) thin films and Rietveld refinement result. Measured diffraction pattern (Y_{obs}) and a calculated diffraction pattern (Y_{calc}) are represented as black and red lines, respectively. The difference between the measured and calculated diffraction patterns ($Y_{obs-calc}$) are indicated by a blue line. The black and red bars provide the calculated Bragg peak positions of the M1 and the rutile phases, respectively.

The fit error (i.e., the difference between the measured and calculated diffraction patterns, $Y_{obs-calc}$) is primarily due to the preferred orientation of the film. We are able to observe multiple diffraction peaks by depositing the films on the Si substrates, but the films still have a preferred orientation, which can be seen by the comparison with a powder diffraction pattern. FIG. R3 shows the normalized diffraction pattern of the pure VO_2 thin film on Si substrate and the ICSD powder diffraction of VO_2 M1 phase. We find that by taking the change in the lattice parameter into account, the peak positions of the thin film match those of the powder diffraction pattern well. Thus, even though there is discrepancy in peak intensities between the calculated powder pattern and the pattern from our textured film, we are able to obtain solid information on the lattice parameters from the refinement exercise.

FIG. R3. XRD pattern (black line) of pure VO₂ thin film on Si substrate and the polycrystalline VO₂ (M1 phase) powder diffraction pattern (green), which shows the peak positions and relative intensities.

(4) The remnant hysteresis observed under the optimal conditions is >4K, which is substantially higher than results observed for the shape memory alloys and indeed this is a matter that deserve further comment. If the “happy accident” makes VO₂ such a perfect material, why then cannot the hysteresis be reduced further. Indeed, the results here are entirely not a record even for VO₂: for example, the following work has noted the almost complete elimination of hysteresis upon co-doping: Miyazaki, K.; Shibuya, K.; Suzuki, M.; Sakai, K.; Fujita, J.-I.; Sawa, A., Chromium–niobium co-doped vanadium dioxide films: Large temperature coefficient of resistance and practically no thermal hysteresis of the metal–insulator transition. AIP Advances 2016, 6 (5), 055012. I suspect the strain induced nucleation of other phases has a lot to do with the remnant hysteresis and ultimately represents a substantial limitation of this approach.

As we described in the manuscript, the “happy accident” aspect of the VO₂ has to do with the fact that because of its unique crystal structure, meeting one lattice compatibility condition, namely $\lambda_2 = 1$ automatically satisfies the cofactor condition. This is not the case with many of the shape memory alloys (SMAs) we have previously worked with. This unusual property of VO₂ thus naturally gives us a recipe for meeting the cofactor condition: by tuning λ_2 to be (close to) 1. A key point of our work here therefore

is to demonstrate the utility of this recipe in a straightforward manner using a single-element-substitution composition spread.

As far as being able to further reduce the hysteresis, there are several things to consider. The absolute value of the minimum hysteresis depends on a variety of factors including materials types (bulk polycrystalline vs. thin films, SMAs vs. ceramics, etc.), as well as the specifics of the measurement methods and conditions (including mechanical load (if any), local strain distribution, etc.) In the previous shape memory alloy work [Zarnetta *et al.*, *Adv. Funct. Mater.* **20**, 1917 (2010), Cui *et al.*, *Nat. Mater.* **5**, 286 (2006)], the focus was always the *relative* change/variation in the hysteresis across varied compositions which allowed us to find the hysteresis minimum compositions. It is important to point out that when we “transfer” the selected compositions of SMA thin films exhibiting (close to) “zero” hysteresis (as determined by monitoring the resistance change) to polycrystalline bulk, the bulk alloys would display reduced hysteresis with exceptional functional fatigue behavior as measured by DSC, but the hysteresis of the bulk samples would never reach zero [Zarnetta *et al.*, *Adv. Funct. Mater.* **20**, 1917 (2010)]. In fact, because of release/absorption of the latent associated with the first-order transition, one generally does not expect bulk SMAs to display identically-zero DSC hysteresis even if the lattice compatibility conditions are met. It is the occurrence of the relative minima which is dictated by the non-linear theory. In the previous work on SMA, the lattice compatible compositions always exhibited minimum hysteresis within each *family* of compositions, but the absolute value of the minimum hysteresis was different from family to family (Ni-Ti-Cu vs. Ni-Ti-Au vs. Ni-Ti-Cu-Pd).

So, it is reasonable to expect the same effect in VO₂-based compounds: that further reduction in hysteresis can be achieved by going to different and in particular multiple element substitution strategies (although multiple elements translate to larger compositional landscapes to map). Indeed, Ni-Ti-Cu-Pd displayed smaller minimal hysteresis than Ni-Ti-Cu. As pointed out by the reviewer, Miyazaki *et al.* have shown that by co-substituting with Cr and Nb, a hysteresis width of as small as 0.6 K can be achieved [Miyazaki, *et al.*, *AIP Advances* **6**, 055012 (2016)]. It is interesting to note that according to reported phase diagrams of VO₂ [see Pouget, J. P. and Launois, H., *J. Phys. Colloques* **37**, C4-49 (1976); and also see FIG. 1 in Strelcov, E. *et al.*, *Nano Letters* **12**, 6198 (2012)], W or Nb substitution leads to a reduction effect and also an increase in the lattice constant, and in comparison, using Cr to substitute V leads to an oxidation effect and also a decrease in the lattice constant. Therefore, it is reasonable to expect that in substituting Cr and Nb simultaneously, Miyazaki *et al.* were able to suppress the volume change (during transition), as compared to the case where only one substitution element is used, and thus this has resulted in a hysteresis width as small as 0.6 K. Previously, in Cui *et al.* [Cui *et al.*, *Nat. Mater.* **5**, 286 (2006)], it is reported that in the case of SMA thin films, volume change had little correlation with reduction of

hysteresis (although this result is counterintuitive and surprising). Perhaps, it is the intrinsically more-brittle nature of ceramic thin films which make them more sensitive to the condition of minimizing the volume change. We now modified the manuscript to refer to Miyazaki et al. and mention this point. In particular, on page 12 of the revised manuscript, we now add:

“It is important to note that our study demonstrates a straightforward route for meeting the cofactor condition using a single-element-substitution composition spread: tuning λ_2 to be (close to) 1; and the minimum thermal hysteresis width (as shown in FIG. 5c 3 K and 5 K for the $V_{1-x}W_xO_2/c-Al_2O_3$ sample and the $V_{1-x}W_xO_2/Si$ sample, respectively) obtained in this single-element-substitution composition spread is not necessarily a universal minimum in doped VO_2 samples. In fact, Miyazaki *et al.* have obtained a hysteresis width as small as 0.6 K using both Cr and Nb to substitute V⁴⁸. According to reported phase diagrams^{38,43} of VO_2 , W or Nb substitution leads to a reduction effect and also an increase in the lattice constant, and in comparison, using Cr to substitute V leads to an oxidation effect and also a decrease in the lattice constant. Therefore, it is reasonable to expect that by substituting V with Cr and Nb simultaneously, the volume change during the phase transformation is suppressed, resulting a smaller minimum hysteresis width as compared to the case where only one substitution element is used.”

(5) Two critical parameters need further elucidation: (a) what happens at faster scan rates and (b) during subsequent cycles? The hysteresis in W-doped VO_2 is known to be scan rate dependent, reflective in some measure, the available nucleation sites. Further work has shown the kinetic asymmetry between the forward and reverse transitions derived from their distinctive origins (Fan, W.; Cao, J.; Seidel, J.; Gu, Y.; Yim, J. W.; Barrett, C.; Yu, K. M.; Ji, J.; Ramesh, R.; Chen, L. Q.; Wu, J. Large Kinetic Asymmetry in the Metal-Insulator Transition Nucleated at Localized and Extended Defects. Phys. Rev. B - Condens. Matter Mater. Phys. 2011, 83 (23), 235102-7.). Lattice strain derived from epitaxial matching strongly alters the formation of twin boundaries and as a result modifies the nucleation probabilities of the M2 and T polymorphs. I expect this explains the differences the authors observe between the films grown on Si and Al_2O_3 .

As the reviewer points out, the hysteresis width in W-substituted VO_2 (and in general, first order structural phase transformations in materials) is known to be scan-rate dependent, and this scan-rate dependence is related to the kinetic asymmetry associated with the structural phase transformation [Fan, W., *et al.*, Phys. Rev. B **83**, 235102 (2011)]. We have now carefully studied the scan-rate dependence of our samples, and a representative result is shown in FIG. R4. Specifically, when the scan is faster, the measured hysteresis width is larger, consistent with typical observation in first order structural phase transformation materials. In this work, a fixed slow scanning rate is used to allow a meaningful

comparison across different substitution levels. The key is clearly to perform consistent measurements for different compositions and samples. Moreover, sudden jumps that are frequently observed in measurements performed on nanobeams [Fan, W., *et al.*, Phys. Rev. B **83**, 235102 (2011)] do not appear in the R-T curves of our thin films. Again, the properties of VO₂ thin films have been found to be very different from that of nanobeams because of the stronger stress effect in samples with reduced dimensions. Our film results are consistent with other thin film reports, and behaviors of nanobeams which are not observed in thin films are beyond the scope of the present work. As we already discussed above, TEM analysis revealed that there are no M2 or T polymorph in our films. Therefore, we believe the difference in transport behaviors between films on sapphire and Si are due to the crystallinity.

FIG. R4. Scan rate dependence of the hysteresis width of the resistance measured from a lightly substituted V_{1-x}W_xO₂ strip ($x \sim 0.9\%$) fabricated on a Si substrate

(6) In Figure 5b, there seems to be no formal metal—insulator transition. The transport data indicate a semiconductor-semiconductor transition. Can the authors explain the strong suppression of the magnitude of the transition and its abruptness (indicating a protracted phase coexistence regime) in these samples (which is not in fact observed for high-quality W-doped VO₂ films and nanobeams). How repeatable are these measurements? Past calorimetry measurements reported by the authors have gone out to thousands of cycles but only singular sweeps are shown here.

The decrease in the overall resistance change and the sharpness of the transition upon W substitution is well-documented in the literature [for example, see Reyes *et al.*, Can. J. Phys. **54**, 408 (1976); Pergament

et al., Thin Solid Films **531**, 572 (2013); Shin *et al.*, Materials Research Bulletin **101**, 287 (2018), and Andreev and Klimov, Physics of the Solid State **61**, 1471 (2019)]. In our study, at $\sim 3.5\%$ substitution, where the transition has lost sharpness, the resistance change is still over one magnitude, which is consistent with what has been revealed in previous studies. In addition, the transition temperature and the hysteresis width can still be identified clearly according to the method illustrated in FIG. S5b.

To the best of our knowledge, there has been no report of W-doped (around 3%) VO₂ thin films whose transitions are as sharp as similarly made undoped VO₂ films. The loss of sharpness we observe upon doping is consistent with previous studies.

The term of MIT is historically adopted for describing a transition in which the electrical conductivity changes dramatically in a relatively narrow temperature range. For example, as shown in FIG. 5b, unlike the typical positive temperature coefficient of metal, the temperature coefficient of the film fabricated on Si is slightly negative in the high-temperature phase. In contrast to that, all the curves shown in FIG. 5a show positive temperature coefficients at high temperatures. In fact, the electronic transport of VO₂ is known to have very little temperature dependence in the high-temperature phase, and the sign of this small coefficient can be either positive or negative and is strongly sample dependent [For example, see the resistance curves in the review articles: Shao, Z. *et al.*, NPG Asia Materials **10**, 581 (2018), and Liu, K. *et al.*, Materials Today, **21**, 875 (2018)]. In addition, although some of sample strips show semiconductor-like behavior, the continuous modification seen here indicates that these changes are essentially due to the same effect, that is, a structural phase transformation between a rutile phase and a monoclinic phase. Therefore, to keep it consistent with previous substitution studies, we still refer to the transition as the MIT. Moreover, in other materials reported to display a MIT, a negative temperature coefficient (so called “non-metallic behavior”) has also been reported. For example, the transition in Ca_{1.9}Sr_{0.1}RuO₄ [Moore, R. G. *et al.* Science **318**, 615 (2007)] is also called a MIT despite a negative temperature coefficient on both sides of the transformation.

As far as our past calorimetry measurements, the only extended (more than 20) measurements carried out by (some of) the authors here are our recent work on additive manufactured bulk samples [Hou *et al.*, Science **366**, 2226 (2019)], and not on earlier Ni-Ti-Cu and Ni-Ti-Cu-Pd studies. For the present thin film work, we have performed 15 thermal cycling and found that change in the hysteresis width is less than 5%, and it does not change beyond 15 cycles.

(7) In a similar context of the above point, many measurements of VO₂ thin films and nanobeams indicate

a distinctive step-like alteration of transport corresponding to switching of individual domains-the extended and broad coexistence regimes observed here are likely due to a complex domain microstructure (can this be imaged?) and the stabilization of various polymorphs. If such texturation is indeed observed and given the substantial degradation of the magnitude of the phase transition, this would negate some of the intended benefits of hysteresis reduction.

In our samples, the measured resistance/conductance is a collective behavior of a 0.2 mm wide strip, and it is a statistical average of the phase transformation of the body of the measured thin film structure. This is distinct from what is observed in nanobeams. As we have discussed extensively already above, there have been mounting evidences of differences in various physical properties between VO₂ thin films and nanobeams. For example, in a review article [Whittaker, Patridge, and Banerjee, *Microscopic and Nanoscale Perspective of the Metal-Insulator Phase Transitions of VO₂: Some New Twists to an Old Tale*, J. Phys. Chem. Lett. **2**, 745–758 (2011)], it is summarized: “Interestingly, although there is no evidence for isolation of a stable M2 intermediate phase for bulk and polycrystalline thin films of VO₂ (as noted above), it does seem apparent that such an intermediate phase mediates the transition from the rutile to the M1 phase in nanostructures where it finds particular stabilization as a result of strong surface-induced stress”.

Specifically, in nanobeams, the transport is sensitive to phase changes in micro-sized structures, and thus distinctive step-like alternation in the resistance curve can be observed. As we mention at several places above, our thin films do not display the behavior of nanobeams, and comparison of our samples with idiosyncrasies of nanobeams is not relevant. In our samples, the XRD results have been used to quantitatively identify the structures of the phases involved in the phase transformation, and as extensively discussed above, TEM images suggest that other intermediate polymorphs are not present as a significant volume fraction.

Further, in many structural phase transformation materials, chemical substitution is expected to introduce local disorders, which are further expected to reduce the transformation sharpness and leads to substantial degradation of the magnitude of the phase transformation without introducing any intermediate phases [for example, Liu *et al.*, Phase-transition temperature suppression to achieve cubic GeTe and high thermoelectric performance by Bi and Mn codoping, PNAS **115**, 5332 (2018)]. Therefore, in general, chemical substitution is expected to introduce disorders but not necessarily result in the emerging of intermediate phases. Broadening of the coexistence regime is certainly not in favor of the perspective of practical applications; however, this broadening does not alter our conclusion here on the close relation between the lattice structure and the hysteresis width.

(8) Have the authors attempted to image domain evolution within these samples? Considerable work on phase contrast microscopy is now available and given the broad transitions, I expect that the diminished hysteresis has much to do with the facilitation of percolative mesoscale textures through this system.

Imaging the evolution of domain dynamics would certainly be of interest to the community, but it is outside the scope of the current work. Our main focus here is to show the correlation between structural compatibility and the minimum hysteresis width, which is summarized in FIG. 5c and 5d. Even if the phase transformation is percolative in nature, we maintain that the prevailing factor governing the hysteresis behavior is lattice compatibility.

(9) As a semantic but nonetheless important point, VO₂ is not a representative electron correlated material given the substantial role of phonons in underpinning the MIT (see for example, Budai, J.D., Hong, J., Manley, M.E., Specht, E.D., Li, C.W., Tischler, J.Z., Abernathy, D.L., Said, A.H., Leu, B.M., Boatner, L.A., McQueeney, R.J., and Delaire, O. (2014) Metallization of vanadium dioxide driven by large phonon entropy. Nature 515, 535–539). As such, extrapolating from VO₂ to Mott systems wherein electron correlation effects are of paramount importance is entirely unjustified and not supported by any of the data presented here.

We agree with the reviewer's comment. The reason for us to use the term 'strongly correlated electron system' for VO₂ is that the material has been historically considered so. We agree that given the fact that phonon plays an important role in the MIT of the material, emphasizing electron correlation as the only dominant effect in this system may be unjustified.

(10) Can the authors provide example of representative WDS data and discuss the accuracy of this quantitation approach. I note that there are no horizontal error bars in Figure 4.

The composition of each sample strip was measured multiple times by WDS using V₂O₅ and W₂O₃ as standards. The average of measured values was then taken to calculate the substitution percentage (x). An estimate of the measurement error has been obtained during this process. The WDS technique has been a routine method for calibrating the chemical composition of materials by many researchers. When multiple measurements are performed on positions with the same chemical composition, the standard deviation is typically below 0.1 %. In our experiments, thanks to the unique combinatorial fabrication

method, we can also estimate the W concentration at any location of the sample simply based on the distance from the location to the edge of the substrate. We found that the values obtained from the WDS measurements are highly consistent with the estimates using the “geometric” method.

FIG. R1 plots the substitution concentration (x in percentile) as a function of a distance from the sample edge on the pure VO_2 side. The error bar is given by the standard deviation of the multiple measurements on each position.

Reviewer #3 (Remarks to the Author):

The authors applied the non-linear theory of martensite to a strongly correlated oxide system $V(1-x)WxO_2$, showing that the hysteresis of the MIT is minimized when the cofactor conditions are satisfied. The minimized hysteresis may feature high reversibility of the material, which is desired in applications.

This work provides a systematic way to reduce the hysteresis of the MIT in W-doped VO_2 and improves our understanding of the hysteresis of the MIT. It will be of interest to the MIT research community.

Below are my comments on this work.

1. The local structure around the W dopant is a tetragonal-like structure [Scientific Reports 2, 466 (2012)], which may serve as nucleation sites at phase transitions. This will have an effect on the hysteresis: more nucleation sites are likely to lead to smaller hysteresis. Compared to the lattice compatibility between phases, what is the role of this effect in regulating the hysteresis?

We agree with the reviewer that the chemical substitution not just alters the lattice constant, but also introduces disorders into the material system. These substitution-induced disordered sites are expected to facilitate the phase transformation, and thus reduce the transition temperature and also drive the hysteresis width to become smaller. The reviewer mentioned previous discovery of forming tetragonal-like structures around W ions, and we agree with the reviewer that this may be one of the reasons to have a smaller hysteresis width upon substitution. However, based on that, increasing the W concentration (i.e., increasing the density of the nucleation sites) is expected to monotonically drive the hysteresis width down, similar to the effect on the transition temperature T_C (for example, this is also seen in Tan, X. *et al.*, Sci. Rep. 2, 466 (2012), where the T_C has been found to linearly decrease as increasing the W concentration). In our measurements, we clearly observed a minimum hysteresis-width composition that corresponds to a substitution level satisfying $\lambda_2 = 1$. Therefore, the hysteresis width reaching the minimum coinciding with the satisfaction of $\lambda_2 = 1$ cannot be simply explained by the increase of nucleation sites upon substitution, and thus we contend that satisfying the lattice compatibility according to the non-linear theory is playing a dominant role here.

2. Intuitively, the more deviation from 1 for $bM1/aT$, the larger lattice incompatibility between phases, and thus the larger hysteresis. This is however not the case according to Fig. 4 and Fig. 5. Can the authors discuss this point? Maybe this is related to the first comment, that is, besides the lattice compatibility aspect, there is another aspect (nucleation sites) that can influence the hysteresis.

As we described in the above reply to the reviewer's first comment, we agree that besides the degree of lattice compatibility between the two end phases, there is another effect which can affect the width of the hysteresis loop. Specifically, as the W concentration increases, the T_C and the hysteresis width are both expected to decrease. However, the decrease in the hysteresis width is expected to be monotonic with increasing the substitution level, which cannot be directly applied to explain the observed minimum (decreasing first and then increasing) of the hysteresis width at a certain substitution concentration.

3. Figure 5(c) shows that the hysteresis of the polycrystalline $V(1-x)WxO_2/Si$ is larger than that of the epitaxial $V(1-x)WxO_2/Al_2O_3$. But in general better crystals should exhibit larger hysteresis due to fewer nucleation sites. Can the authors give a more detailed explanation for the measured result?

In addition to the substitution-induced lattice compatibility, there are other factors which can influence the hysteresis width measured here by the resistive transition. For instance, grain boundaries might be playing a role in that adjacent epitaxial grains might have a degree of coherence making them conducive to transforming together, as opposed to textured grains in polycrystalline films where grain boundaries can be detrimental to such a transformation mode. Our results here that undoped VO_2 we made on Al_2O_3 show smaller hysteresis than the films on Si substrates (FIG. 5a and 5b) are consistent with a previous study [Xiong *et al.*, J. Phys. D: Appl. Phys. **47**, 455304 (2014)], where VO_2 films fabricated on Si substrates with an Al_2O_3 layer as a buffer showed a much narrower transition width as compared to films directly fabricated on Si substrates.

Reviewers' comments:

Reviewer #1 (Remarks to the Author):

The authors have addressed all my concerns and I recommend publication.

Reviewer #2 (Remarks to the Author):

The authors have undertaken more extensive characterization of the prepared films to rule out the co-existence of M2 and triclinic polymorphs. I remain somewhat perplexed by some of the rationale that has been put forth, specifically the idea that surface stresses in free-standing nanobeams are more likely than residual stresses in vapor-deposited thin films –surely facile deformation of the former provides a means of relaxing such stresses. Nevertheless, under the specific conditions examined here, the authors make a compelling case for there not being a significant concentration of the M2 phase. Since hysteretic phenomena have been studied extensively in nanobeams, I suggest that the authors provide an expanded discussion clearly delineating the differences between the observations here and previous studies of the origin of hysteresis in VO₂ nanobeams. Such a discussion would allow this work to better connect to the considerable literature on nanobeams and would initiate a much needed discussion as the role not just of stresses but other parameters such as oxygen stoichiometry and point defect concentration.

In a similar vein to the above point, by contrasting the lattice parameters extracted from Rietveld refinements for undoped VO₂ films, can the authors delineate how much strain there might be in the films on Si and sapphire?

The scan rate dependence of hysteresis widths is quite remarkable (hysteresis increases to 24C at faster scan rates) and points to the role in correlated oxides of not just thermodynamics but also activation barriers, which likely have some electronic structure contributions. Figure R4 is quite useful and it would be nice if the authors could more explicitly comment on the kinetic aspects of the transition. I am not asking for more experiments but just a more detailed discussion. There are nuances here that are clearly not well understood that ought to be highlighted and not entirely relegated to the SI.

The authors make a compelling point about the role of disorder in suppressing the magnitude of the MIT. An alternative view comes from the idea of the orbital-selective Mott transition in VO₂ (e.g., J. Appl. Phys. 125, 082539, (2019)) in epitaxial VO₂ films that seem rather similar to the ones analyzed here wherein the authors find that even before the onset of the Peirl's instability, the residual stresses engender breaking of the degeneracy of the electronic states, thereby inducing an orbital-selective Mott transition (OSMT). Such an OSMT is then inevitably lower in magnitude than a full coupled Mott-Peirls transition and provides a more physical rationale for the suppressed transition. The work I am citing further provides notable caveats about the generalizability of the OSMT to other correlated oxides. This again gets to the crux of my critique that co-factor analysis and single-element substitution appears to work under a specific set of circumstances (that enable relaxation of interfacial stresses in a particular manner and under certain density of defects where the transitions are not nucleation limited) but electronic structure matters and substantially confounds hysteretic phenomena in correlated systems. As such, some tempering of the claims regarding broad applicability of crystallographic analysis-that ultimately alters only phonon density of states- to systems with strongly coupled lattice, electronic, and spin degrees of freedom is necessary.

I have a further question about the lattice compatibility analysis with seeking to get to a co-factor of two. Unlike in previous systems subject to lattice compatibility conditions, tungsten is an aliovalent dopant that must inevitably be charge compensated by localized vanadium reduction or an oxygen interstitial (it is likely polaron distortion on vanadium centers that dominates). Nevertheless, an important idea for why co-factor analysis works here –amplifying an argument that Reviewer 3 seems

to make-is that the local distortion/polaron formation likely yields an abundance of nucleation sites that at a minimum ensure that at low scan rates the phase transitions are not nucleation limited. This might be a second happy accident for tungsten that facilitates observation of this phenomenon since a homovalent dopant that meets lattice compatibility conditions might still have remained nucleation limited across the entire range of scan rates.

Reviewer #3 (Remarks to the Author):

Now I recommend this manuscript for being published on Nature Communications. Comment: I was not stating that the disorder aspect is dominant compared to the lattice-compatibility aspect, but my opinion is that adding some discussion about the disorder effect on the hysteresis may make the manuscript more complete.

Reviewer #2 (Remarks to the Author):

The authors have undertaken more extensive characterization of the prepared films to rule out the co-existence of M2 and triclinic polymorphs. I remain somewhat perplexed by some of the rationale that has been put forth, specifically the idea that surface stresses in free-standing nanobeams beams are more likely than residual stresses in vapor-deposited thin films –surely facile deformation of the former provides a means of relaxing such stresses. Nevertheless, under the specific conditions examined here, the authors make a compelling case for there not being a significant concentration of the M2 phase. Since hysteretic phenomena have been studied extensively in nanobeams, I suggest that the authors provide an expanded discussion clearly delineating the differences between the observations here and previous studies of the origin of hysteresis in VO₂ nanobeams. Such a discussion would allow this work to better connect to the considerable literature on nanobeams and would initiate a much needed discussion as the role not just of stresses but other parameters such as oxygen stoichiometry and point defect concentration.

As acknowledged by the Referee above, we have already established that thin films are different from nanobeams and that our films do not have the M2 phase. As far as the statement regarding the surface stresses in free-standing nanobeams, we refer to a review article by Whittaker, Patridge, and Banerjee [J. Phys. Chem. Lett. **2**, 745 (2011)], where they point out “inevitable lattice stresses introduced as a result of both intrinsic geometric effects of high surface-to-volume ratios of nanostructures as well as extrinsic strong substrate interactions portend modifications of the thermodynamic stabilities of the different phases.” In other words, in free-standing nanobeams, stabilities of materials phases are inevitably altered due to the intrinsic high surface-to-volume ratio.

Specifically, experimental studies on VO₂ nanobeams having weak interaction with the underneath substrates have confirmed the presence of the M2 phase through the MIT [Zhang, *et al.*, Nano Lett. **9**, 4527 (2009); Sohn, *et al.*, Nano Lett. **9**, 3392 (2009)], which is clearly different from what is usually seen in VO₂ thin films. In fact, it is not surprising at all that such a geometric effect further modifies the phase diagram of VO₂ and alters the phase-transition temperatures and hysteresis [Whittaker, *et al.*, J. Phys. Chem. Lett. **2**, 745 (2011)]. As we previously stated, the observed differences between nanobeams and thin films have been explained extensively in the literature. In the revised manuscript, we describe the reported differences between nanobeams and thin films, and then again point out how the absence of the M2 phase in our thin films is not surprising at all.

In the revised manuscript, the last paragraph on page 8 now reads:

“It has been found that the intermediate M2 phase of VO₂ appears often in nanobeams³⁹ and sometimes in thin films^{40,41}. The fact that the M2 phase is more prevalent in nanobeams is presumably due to the stronger geometric effect in nanobeams⁴². Specifically, the M2 phase has been observed even in nanobeams where interfacial coupling between the nano-sized materials and the underneath substrates is negligible^{43,44}, suggesting that the intrinsic geometric effects due to the high surface-to-volume ratios of these quasi-1-dimensional structures facilitate the formation of the intermediate phase through the MIT. Further, according to a number of previous studies^{39,41,45,46}, replacing V ions with a small amount (up to few %) of W is expected to reduce the transition temperature of the MIT without introducing the M2 phase through the structural transformation. Therefore, the absence of evidence of the M2 phase in our composition-spread films is consistent with previously V_{1-x}W_xO₂ work on *thin films* of V_{1-x}W_xO₂, where unlike in nanobeams, the M2 phase was not observed.”

We have now also added a discussion on the transformation and the hysteresis in VO₂ in general (including nanobeams) in the Discussion section on page 13.

“The first order transition in VO₂ is a complex process involving change in the electronic structure as well as transformation of the crystal structure accompanied by a large latent heat. The subtle interplay between various factors is reflected in the sensitivity of the transformation temperature and its hysteresis to small chemical modifications and geometrical effects (e.g. films versus nanobeams) including strain. The fact that lattice compatibility has influence on the hysteresis as observed here underscores the basic role crystal structure plays in determining the physical properties of materials in general.”

In a similar vein to the above point, by contrasting the lattice parameters extracted from Rietveld refinements for undoped VO₂ films, can the authors delineate how much strain there might be in the films on Si and sapphire?

The question is whether any residual strain in the films has any influence on the central results reported here. The polycrystalline nature of the VO₂ films on the Si substrate implies that the strain from the substrate is minimal because there is no epitaxial strain. For the epitaxial films on sapphire also, because the films are sufficiently thick (150 nm), the strain is fully relaxed which is reflected in the fact that the VO₂ (020) peak position (39.82°) of the film is very close to that of bulk (39.69°). From our previous work on lattice compatibility of shape memory alloy thin films with similarly fully-relaxed thick films on Si substrates, small residual strain in relaxed films was found not to affect the trend of the hysteresis

reduction as a function of lattice constants [Please see: Robert Zarnetta, *et al.*, *Advanced Functional Materials* **20**, 1917 (2010)].

The scan rate dependence of hysteresis widths is quite remarkable (hysteresis increases to 24C at faster scan rates) and points to the role in correlated oxides of not just thermodynamics but also activation barriers, which likely have some electronic structure contributions. Figure R4 is quite useful and it would be nice if the authors could more explicitly comment on the kinetic aspects of the transition. I am not asking for more experiments but just a more detailed discussion. There are nuances here that are clearly not well understood that ought to be highlighted and not entirely relegated to the SI.

Data in Figure R4 in our previous reply were measured and plotted upon the request of the Reviewer. The observed scan rate dependence of the hysteresis is a well-known phenomenon for first order transitions with latent heat in general (as observed in any differential scanning calorimetry measurements of martensitic transformation), and there is no reason to believe that it is related to electronic structure or kinetics of the transition, especially for thin films where measurements reflect the collective behavior of a large number of microstructures, such as grains, grain boundaries, nucleation sites, etc. in a statistically averaged manner. As we already discussed in the last reply, what this plot underscores is the importance of keeping the consistent and slow scan rate for study of a first-order transition accompanied by a large latent heat as in the case VO₂. A constant scan rate has allowed us to make a meaningful comparison across the entire doping range of the sample.

*The authors make a compelling point about the role of disorder in suppressing the magnitude of the MIT. An alternative view comes from the idea of the orbital-selective Mott transition in VO₂ (e.g., *J. Appl. Phys.* **125**, 082539, (2019)) in epitaxial VO₂ films that seem rather similar to the ones analyzed here wherein the authors find that even before the onset of the Peirl's instability, the residual stresses engender breaking of the degeneracy of the electronic states, thereby inducing an orbital-selective Mott transition (OSMT). Such an OSMT is then inevitably lower in magnitude than a full coupled Mott-Peirls transition and provides a more physical rationale for the suppressed transition. The work I am citing further provides notable caveats about the generalizability of the OSMT to other correlated oxides. This again gets to the crux of my critique that co-factor analysis and single-element substitution appears to work under a specific set of circumstances (that enable relaxation of interfacial stresses in a particular manner and under certain density of defects where the transitions are not nucleation limited) but electronic structure matters and substantially confounds hysteretic phenomena in correlated systems. As such, some tempering of the claims regarding broad applicability of crystallographic analysis-that*

ultimately alters only phonon density of states- to systems with strongly coupled lattice, electronic, and spin degrees of freedom is necessary.

The theoretical work cited by the reviewer proposes a change in the electronic structure of VO₂ system when strain is involved. The Reviewer further suggests that this OSMT leads to lowering the transition magnitude and has more ‘generalizability’ for correlated oxides than the co-factor analysis provided in our manuscript. We understand that when W is introduced to replace V, internal stress may be introduced, and thus the OSMT cannot be ruled out even before the system undergoes the structural phase transformation. However, the proposed theory cannot explain our observation of systematic doping leading to a cusp/minimum in the hysteresis *and* simultaneously the middle eigenvalue of the transformation matrix approaching unity. Our conclusion therefore is that OSMT is not playing a major role here.

As far as the crux of the reviewer’s critique, it is not our intention to claim that lattice compatibility explains and supersedes *all* aspects of the materials with respect to transformation and the hysteresis, especially in a complex correlated materials system as VO₂. At the same time, we have made a compelling case of its influence on the transformation here. As we stated above (and added in a new paragraph in Discussion), given that structure and lattice parameters are a fundamental attribute of any crystalline material, the observation is reasonable, as the Reviewer agrees. Considering that there are other factors which are related to reduction of the transition temperature and the transition sharpness, we have now revised a sentence to “temper” and better describe the implication of our observation to on-going research of transformation materials at large. Specifically, the last two sentences in the discussion section on page 13 are changed to:

“The success of applying the non-linear theory to identify the conditions for the ultra-compatible MIT in a functional oxide system suggests that the theory is highly valuable in guiding the optimization of transforming materials. Moreover, the correlation between the MIT and the crystal structure demonstrated in our study also suggests that the structural phase transformation plays a central role in the observed MIT in the V_{1-x}W_xO₂ system.”

I have a further question about the lattice compatibility analysis with seeking to get to a co-factor of two. Unlike in previous systems subject to lattice compatibility conditions, tungsten is an aliovalent dopant that must inevitably be charge compensated by localized vanadium reduction or an oxygen interstitial (it is likely polaron distortion on vanadium centers that dominates). Nevertheless, an important idea for why co-factor analysis works here –amplifying an argument that Reviewer 3 seems to make-is that the local

distortion/polaron formation likely yields an abundance of nucleation sites that at a minimum ensure that at low scan rates the phase transitions are not nucleation limited. This might be a second happy accident for tungsten that facilitates observation of this phenomenon since a homovalent dopant that meets lattice compatibility conditions might still have remained nucleation limited across the entire range of scan rates.

As we described in our previous reply to Reviewer 3's comment, introducing W certainly can introduce more disorders (which may act as nucleation sites) into the system, and that is the reason that the transition temperature shows a monotonic shift to lower temperature as the W concentration increases. However, the change in the disorder density upon doping alone cannot explain the observed minimum in the hysteresis width at a certain W concentration.